# Dissecting apicoplast functions through continuous cultivation of *Toxoplasma gondii* devoid of the organelle

Min Chen[1], Szilamér Gyula Koszti[1], Alessandro Bonavoglia[1], Bohumil Maco[1], Olivier von Rohr[1], Hong-Juan Peng[2]✉, Dominique Soldati-Favre[1]✉ & Joachim Kloehn[1]✉

The apicoplast, a relic plastid organelle derived from secondary endosymbiosis, is crucial for many medically relevant Apicomplexa. While it no longer performs photosynthesis, the organelle retains several essential metabolic pathways. In this study, we examine the four primary metabolic pathways in the *Toxoplasma gondii* apicoplast, along with an accessory pathway, and identify conditions that can bypass these. Contrary to the prevailing view that the apicoplast is indispensable for *T. gondii*, we demonstrate that bypassing all pathways renders the apicoplast non-essential. We further show that *T. gondii* lacking an apicoplast (*T. gondii*[−Apico]) can be maintained indefinitely in culture, establishing a unique model to study the functions of this organelle. Through comprehensive metabolomic, transcriptomic, and proteomic analyses of *T. gondii*[−Apico] we uncover significant adaptation mechanisms following loss of the organelle and identify numerous putative apicoplast proteins revealed by their decreased abundance in *T. gondii*[−Apico]. Moreover, *T. gondii*[−Apico] parasites exhibit reduced sensitivity to apicoplast targeting compounds, providing a valuable tool for discovering new drugs acting on the organelle. The capability to culture *T. gondii* without its plastid offers new avenues for exploring apicoplast biology and developing novel therapeutic strategies against apicomplexan parasites.

Several apicomplexan parasites, including *Plasmodium falciparum* and *Toxoplasma gondii*, pose a severe threat to human health[1–3]. Others, such as *Neospora* and *Theileria* spp., inflict considerable economic losses by infecting livestock[4,5]. A hallmark of many apicomplexans is the presence of the apicoplast, an organelle derived from secondary endosymbiosis of a red alga, that has lost its photosynthetic function but harbors vital anabolic pathways[6,7]. Apicomplexans that possess an apicoplast, like *Plasmodium* spp. and *T. gondii*, are critically dependent on it. The organelle maintains its own circular genome[8–10], encoding

tRNAs and rRNAs but only 28 proteins while the majority of apicoplast proteins (estimated to be 300-500 proteins[11]) are nuclear encoded and imported into the organelle[12,13]. Since its discovery approximately three decades ago, the apicoplast's maintenance and its metabolic pathways have been regarded as prime drug targets due to its evolutionary divergence and critical role for parasite survival[14]. To date, various compounds have been identified to target the apicoplast[15]. Disruption of apicoplast maintenance results in the organelle's loss, leading to parasite death during the second lytic cycle, a phenomenon

[1]Department of Microbiology and Molecular Medicine, University of Geneva, CMU, Geneva, Switzerland. [2]Department of Pathogen Biology, Guangdong Provincial Key Laboratory of Tropical Diseases Research, School of Public Health; Key Laboratory of Infectious Diseases Research in South China (Ministry of Education), Southern Medical University, Guangzhou City, Guangdong Province, China. ✉e-mail: Floriapeng@hotmail.com; Dominique.soldati-favre@unige.ch; Joachim.Kloehn@unige.ch

known as delayed death[16], while inhibition of some apicoplast functions causes a favourable rapid death[15,17].

More than a decade ago, a landmark study demonstrated that the intraerythrocytic stage of *P. falciparum* can survive apicoplast loss if supplemented with a single metabolite, the isoprenoid precursor isopentenyl pyrophosphate (IPP), in the culture medium[18]. This study uncovered the apicoplast's limited function in intraerythrocytic *P. falciparum* and provided an invaluable tool for studying the organelle. Specifically, uncoupling apicoplast-loss from parasite survival facilitated the characterization of apicoplast proteins[19,20]. In addition, drug screens comparing the efficacy of compounds against *P. falciparum*, in the presence or absence of IPP, provide a simple means to identify apicoplast-targeting compounds, revealed by their reduced efficacy in the presence of IPP[21–27]. Bypassing the parasite's endogenous IPP synthesis pathway was later refined through the genetic complementation with a four-gene mevalonate (MVA) bypass cassette, enabling parasites to synthesize IPP from mevalonolactone (MVL), a cost-effective alternative to the expensive IPP[28].

Although the ability to culture *P. falciparum* following loss of the organelle substantially advanced research into the organelle's functions and potential inhibitors, the full exploitation of this tool is limited by the restricted genetic tractability of *P. falciparum* and the apicoplast's reduced roles in blood stage malaria parasites[29,30].

In contrast to intraerythrocytic *P. falciparum*, *T. gondii* tachyzoites rely on at least four metabolic pathways hosted within the apicoplast: (1) de novo fatty acid (FA) biosynthesis via a FASII pathway[31,32], (2) synthesis of lysophosphatidic acid (LPA), a lipid precursor[33], (3) parts of the heme synthesis pathway[34,35] and (4) synthesis of the isoprenoid precursor IPP via the so-called non-MVA or 2-C-methyl-D-erythritol 4-phosphate/1-deoxy-D-xylulose 5-phosphate pathway (MEP/DOXP) pathway[36–38]. The metabolites generated by these pathways are critically needed for cellular functions outside of the apicoplast, such as membrane biogenesis, mitochondrial respiration, and protein trafficking[35,39–42]. In contrast to these output-generating pathways, other apicoplast-resident pathways, including its housekeeping functions, such as replication, transcription and translation, as well as the synthesis of lipoate, iron-sulfur clusters and reducing power[36,43,44], function as supporting pathways. These do not produce direct metabolic outputs but instead, create a suitable environment and supply essential factors to sustain the output-generating pathways. Recent studies have shown that the *T. gondii* apicoplast output-generating pathways can, at least in part, be bypassed through metabolite supplementations[31,32,33,45].

Here, we set out to test this systematically. To this end, we targeted enzymes in each apicoplast-resident output-generating pathway, as well as a supporting pathway, and partially or fully rescued their functions through metabolite supplementations and genetic complementation. Next, we demonstrate, for the first time, that bypassing all output-generating pathways can sustain *T. gondii* parasites that have permanently lost the apicoplast. We demonstrate that this bypass can be used to identify apicoplast-targeting drugs, including drugs acting on pathways that are dispensable for blood-stage *P. falciparum* but fitness conferring for *T. gondii*. In addition, we meticulously characterize apicoplast-less parasites (*T. gondii*⁻ᴬᵖⁱᶜᵒ) and leverage this tool to study the organelle's functions, ultimately leading to the discovery of a novel critical coccidian-specific apicoplast protein.

## Results

### Direct partial rescue of the apicoplast's fatty acid and lipid precursor synthesis pathways

The *T. gondii* apicoplast harbors four metabolic pathways that produce essential metabolites needed for parasite survival and replication (Fig. 1)[46]. In addition, several pathways operate to support these output-generating pathways by providing cofactors and substrates.

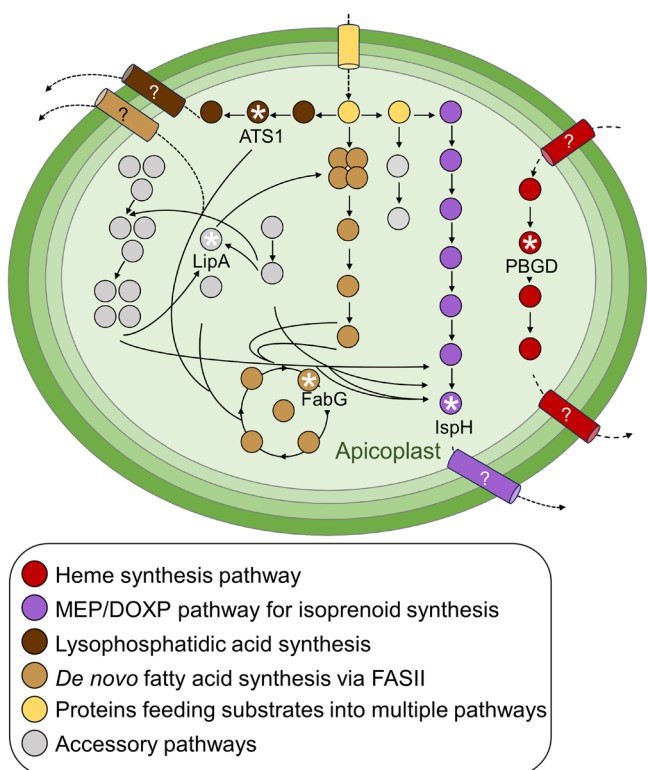

**Fig. 1 | Schematic overview of the metabolic pathways in the apicoplast of *T. gondii*.** Schematic depiction of the four-membrane enclosed apicoplast organelle. Enzymes are shown as circles, and transporters as barrels. Note that, for simplicity, transporters are depicted spanning multiple membranes. Enzymes of output-generating pathways that produce metabolites needed outside of the apicoplast are shown as colored circles (see legend), while supporting pathway enzymes are depicted in gray. Products/intermediates of the isoprenoid and heme synthesis pathways are exported and further processed in the cytosol or at the mitochondrion, while fatty acids and lipids generated in the apicoplast can be further processed at the endoplasmic reticulum (not shown). Enzymes investigated as part of this study are highlighted by their provided abbreviation and are marked with a white asterisk. Expected transporters of unknown identity are indicated by question marks. Abbreviations: PBGD, porphobilinogen deaminase; IspH, 4-hydroxy-3-methylbut−2-enyl diphosphate reductase; FabG, 3-ketoacyl-acyl carrier protein reductase; LipA, lipoic acid synthase; ATS1, glycerol 3-phosphate acyl-transferase 1; MEP/DOXP, 2-C-methyl-D-erythritol 4-phosphate/ 1-deoxy-D-xylulose 5-phosphate; FASII, type II fatty acid synthase.

Legend:
- Heme synthesis pathway
- MEP/DOXP pathway for isoprenoid synthesis
- Lysophosphatidic acid synthesis
- *De novo* fatty acid synthesis via FASII
- Proteins feeding substrates into multiple pathways
- Accessory pathways

Several studies have indicated a complete or partial rescue of some apicoplast output generating pathways through metabolite supplementations[31,32,33,45]. To test this systematically, we generated conditional mutants of a protein in each output-generating pathway, as well as in one supporting pathway. First, we targeted enzymes in the apicoplast-resident fatty acid (FA) and lipid precursor synthesis pathways: specifically, the FASII component, 3-ketoacyl-acyl carrier protein (ACP)-reductase (FabG), the plant-like glycerol 3-phosphate acyltransferase, (ATS1)[33] and the lipoic acid synthase (LipA). These directly (FabG, ATS1) or indirectly (LipA) participate in the synthesis of FAs or the related lipid precursor lysophosphatidic acid (LPA). All mutants were generated using CRISPR-Cas9 editing[47,48], in a parasite line that expresses dimerizable Cre recombinase (DiCre)[49]. A 3-Ty epitope tag was fused to the endogenous locus and *LoxP* sites were simultaneously inserted after the stop codon, followed by a U1 recognition site in the 3' untranslated region[49] (Supplementary Fig. 1a). In addition, a hypoxanthine-xanthine-guanine phosphoribosyl transferase (*HXGPRT*) resistance cassette was inserted (Supplementary Fig. 1a), facilitating the selection of positive transfectants, by treatment with mycophenolic acid and xanthine[50], followed by limited dilution cloning.

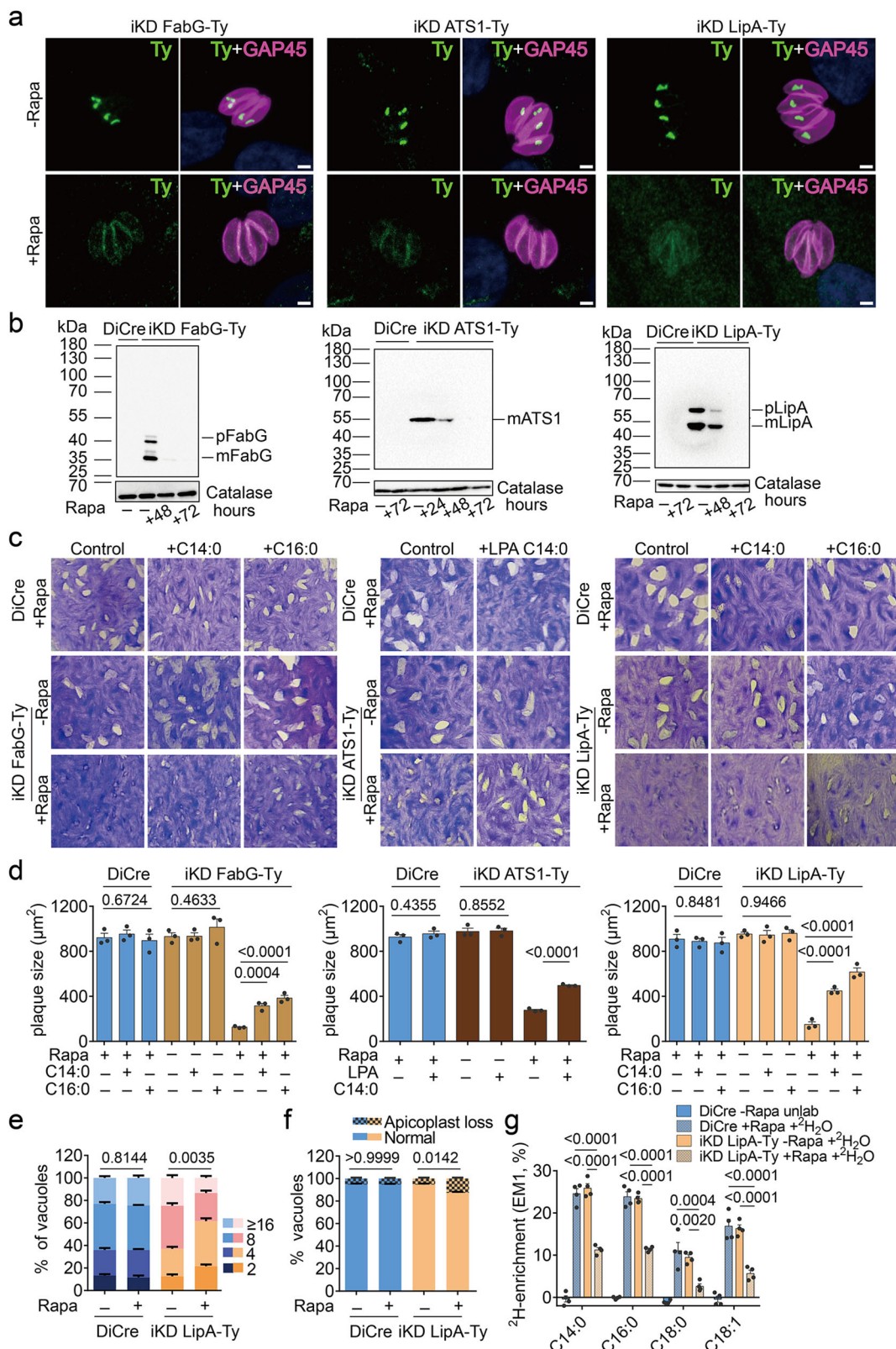

Integration of the above constructs at the endogenous locus of FabG, ATS1 and LipA was validated by genomic PCR (Supplementary Fig. 1b), using the primers listed in Supplementary Data 1. As expected, the epitope-tagged enzymes localized to the apicoplast, based on their co-localization with the apicoplast lumen marker chaperonin 60 (Cpn60)[51] in indirect immunofluorescence assays (IFAs) (Supplementary Fig. 1c). Upon addition of rapamycin the 3' flanking sequences

between the *LoxP* sites were excised, leading to mRNA degradation and efficient downregulation of the selected proteins, as assessed by IFAs (Fig. 2a) and western blots (Fig. 2b). Proteins were depleted below the level of detection within 48–72 h of rapamycin-treatment (Fig. 2b). The pre-processed (p) and the mature (m) epitope-tagged proteins ran close to their predicted molecular weights (pFabG - 46 kDa, mFabG - 35 kDa; mATS1 - 55 kDa; pLipA - 65 kDa, mLipA - 55 kDa) (Fig. 2b).

**Fig. 2 | Partial rescue of the fatty acid and lipid precursor synthesis pathways in the *T. gondii* apicoplast. a** Indirect immunofluorescence assays (IFAs) of iKD FabG-Ty, iKD ATS1-Ty and iKD LipA-Ty parasites treated with rapamycin (Rapa) for 72 h or not, stained with anti-Ty, and anti-GAP45 (pellicle marker) antibodies ($n=3$). **b** Western blots of iKD FabG-Ty, iKD ATS1-Ty, and iKD LipA-Ty and its parental controls after varying durations of Rapa treatment. Membranes were probed with anti-Ty and anti-catalase (loading control) antibodies ($n=3$). **c, d** Plaque assays of DiCre and iKD FabG-Ty, iKD ATS1-Ty and iKD LipA-Ty in normal medium or medium supplemented with 100 μM myristic (C14:0) or palmitic acid (C16:0) (iKD FabG-Ty and iKD LipA-Ty) or 20 μM lysophosphatidic acid (LPA 0:0/14:0) (iKD ATS1-Ty) **(c)** and quantification of plaque sizes ($n=3$) **(d)**. **e** Intracellular growth assay of DiCre and iKD LipA-Ty after 24 h of intracellular growth and treatment for a total of 72 h

with Rapa or not ($n=3$). **f** Quantification of apicoplast loss of iKD LipA-Ty following treatment as for e ($n=3$). **g** Measurement of $^2$H incorporation into major fatty acids of DiCre and iKD LipA-Ty parasites following 72 h of Rapa treatment and 24 h of $^2$H$_2$O labeling (3.6% v/v) as measured by GC-MS ($n=4$). Pictures in (**a**, **b**, and **c**) are representative of 3 independent experiments. Bar graphs in (**d–g**) show the means (bars) of three or four independent experiments. In (**d** and **g**), dots represent the means of each experiment, based on at least 10 plaques and 4 biological replicates, respectively. Error bars indicate the standard deviation. In (**d–f**), Student's two-sided *t* tests compare the indicated conditions. In (**g**), the indicated conditions were compared via a One-way ANOVA followed by Tukey's pairwise comparison. All *p*-values are given and were considered significant at $p < 0.05$. Scale bars in a: 2 μm. Source data are provided as a Source Data file.

Downregulation of each protein was associated with the formation of significantly smaller plaques in lysis plaque assays (Fig. 2c, d). However, consistent with previous reports[31,32,33], exogenous supplementation of the corresponding metabolites (FA C14:0, C16:0 conjugated to BSA or LPA 0:0/14:0) could partially rescue the plaque assay defect associated with the disruption of the respective pathway (Fig. 2c, d). Concordant with the rescuing effect of exogenous FAs and lipids, the plaque assay defect was most pronounced when parasites were cultured in low serum (1%) and partially alleviated in higher serum concentrations (Supplementary Fig. 2a–e).

As previously described for ATS1 and several FASII components, the downregulation of FabG was associated with partial apicoplast loss[31,32,33], as determined by loss of signal for the Trx-like protein 1 (ATrx1, apicoplast membrane marker) (Supplementary Fig. 2f)[52]. Partial apicoplast-loss was further validated by employing Cpn60 and DAPI staining (apicoplast genome) (Supplementary Fig. 2g). Comparable rates of apicoplast loss were observed with each marker (Supplementary Fig. 2h), leading us to standardize apicoplast-loss quantification based on a single marker, ATrx1, in subsequent analyses. In contrast to the plaque assay defect, apicoplast loss was not effectively rescued by the supplementation of exogenous FAs (Supplementary Fig. 2i).

While several FASII components and ATS1 have previously been investigated, LipA has, to our knowledge, not been targeted genetically in *T. gondii*, prompting us to investigate its function in greater depth. Together with LipB, LipA participates in the synthesis of lipoic acid, needed for the lipoylation, a critical posttranslational modification, of pyruvate dehydrogenase (PDH)[32,43,53]. Downregulation of LipA was associated with an intracellular growth defect (Fig. 2e) but with only modest apicoplast loss (Fig. 2f). To further validate the role of LipA in supporting FA synthesis, we employed heavy water ($^2$H$_2$O, 3.6% v/v) labeling followed by gas chromatography-mass spectrometry, (GC-MS). In the presence of $^2$H$_2$O, deuterium ($^2$H) atoms are incorporated into newly formed FAs during several steps of de novo synthesis but not into pre-existing FAs[54,55]. Thus, $^2$H incorporation provides a measure for the rate of FA synthesis. GC-MS analyses revealed reduced incorporation of $^2$H into FAs of parasites depleted in LipA, consistent with a reduced synthesis rate (Supplementary Data 2 and Fig. 2g). Collectively, these results reveal that the apicoplast-resident FA and lipid precursor synthesis (LPA), as well as their supporting pathways, can be partially rescued through direct provision of their respective end-product.

### Indirect bypass of apicoplast-resident heme and isoprenoid synthesis in *T. gondii*

In *T. gondii*, the apicoplast also harbors parts of the heme synthesis pathway as well as the complete MEP/DOXP pathway for the synthesis of the isoprenoid precursor IPP (Fig. 1). To scrutinize these pathways, we generated conditional mutants for the apicoplast-resident enzymes porphobilinogen deaminase (PBGD) and the 1-hydroxy-2-methyl-2-(E)-butenyl 4-diphosphate reductase (IspH or LytB), acting in the heme synthesis and the MEP/DOXP pathway, respectively. The loci were

manipulated as described above and the generation of the desired transgenic strains was confirmed by genomic PCR (Supplementary Fig. 3a), using the primers listed in Supplementary Data 1. Both enzymes were localized to the apicoplast (Supplementary Fig. 3b), consistent with previous reports[38,56] and could be efficiently down-regulated below the level of detection within 48–72 h as observed by IFA (Fig. 3a) and western blot (Fig. 3b). Epitope-tagged pre-processed and mature PBGD and IspH ran close to their predicted molecular weights (pPBGD - 85 kDa, mPBGD - 70 kDa; pIspH - 83 kDa, mIspH - 55 kDa) (Fig. 3b). Downregulation of both proteins caused a significant growth defect (Supplementary Fig. 3c, d) but no noticeable apicoplast loss (Supplementary Fig. 3e, f). The downregulation of PBGD was associated with a significant reduction in plaque sizes, that could partially be rescued by the provision of exogenous 5-aminolevulinic acid (5ALA) (Fig. 3c, d). This aligns with our previous observation that exogenous 5ALA can partially alleviate defects in the parasite's heme synthesis pathway, presumably by upregulating heme synthesis inside the host and boosting the parasite's ability to salvage heme or its precursors[31,34].

Next, we assessed the importance of the MEP/DOXP pathway and the ability to bypass it. Since *T. gondii* is unable to salvage exogenous IPP due to its charged nature[36], we opted to insert the HA-tagged mevalonate bypass cassette (MVA)[28,45] into the uracil phosphoribosyl transferase (*UPRT*) locus of iKD IspH-Ty parasites[57], generating an iKD IspH-MVA-HA line. Integration of the MVA cassette leads to the expression of a multifunctional polypeptide of human and bacterial origin that converts mevalonolactone (MVL) to IPP/DMAPP[28]. Integration of the MVA cassette was confirmed by PCR (Supplementary Fig. 3g). The HA-tagged polyprotein was detected in the cytosol of iKD IspH-MVA-HA by IFA (Fig. 3e) and in parasite extracts by western blot, running close to the expected molecular weight of about 113 kDa (Fig. 3f). As expected, depletion of IspH, under standard culture conditions, abolished the formation of plaques (Fig. 3g, h). However, remarkably, the provision of high levels of MVL (10 mM) fully restored the plaque assay defect in parasites depleted in IspH that express the MVA cassette (Fig. 3g, h). Lower levels of MVL only supported the formation of smaller plaques (5 mM) or no plaques (1 mM) (Supplementary Fig. 3h, i). Together, these results reveal that each apicoplast-resident output-generating and its associated supporting pathways can be partially rescued or fully bypassed through a genetic modification and metabolite supplementations.

### Continuous culture and phenotypic characterization of apicoplast-less *T. gondii*

Having bypassed each apicoplast output-generating pathway individually, we next aimed to determine if the entire organelle could be rendered dispensable, allowing for the continuous cultivation of apicoplast-less *T. gondii* (*T. gondii*$^{-Apico}$). To disrupt the apicoplast, we treated *T. gondii* with actinonin, which was previously shown to inhibit the apicoplast membrane-associated protease FtsH1, a homolog of a bacterial membrane AAA+ metalloprotease[58,59], causing rapid loss of the organelle and subsequent parasite death. Indeed, actinonin-

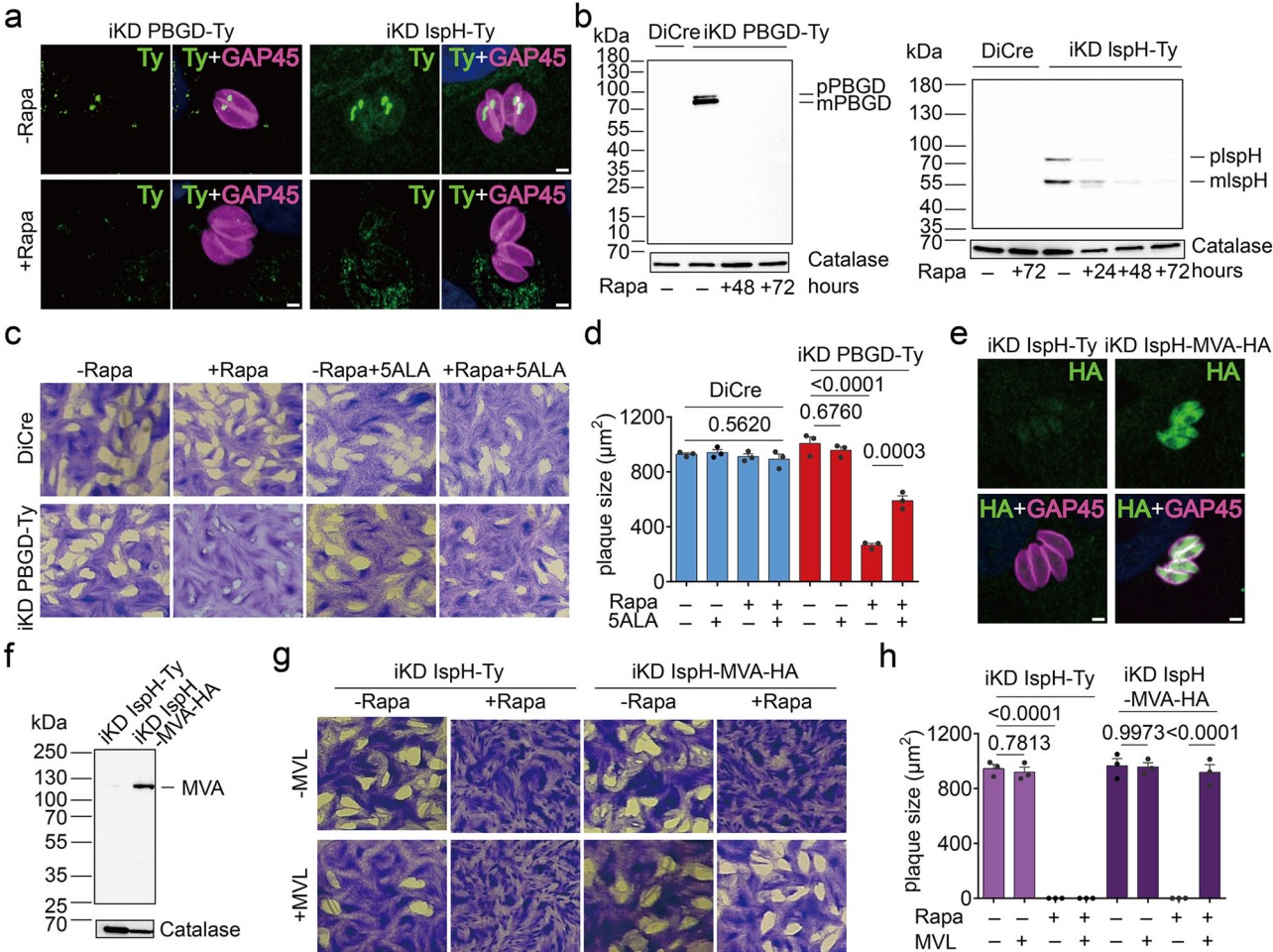

**Fig. 3 | Bypass of the heme and isoprenoid synthesis pathways in the *T. gondii* apicoplast. a** Indirect immunofluorescence assays (IFAs) of iKD PBGD-Ty and iKD-IspH-Ty parasites in the presence (72 h) or absence of rapamycin (Rapa), stained with anti-Ty, and anti-GAP45 (pellicle marker) antibodies (*n* = 3). **b** Western blot of iKD PBGD-Ty and iKD-IspH-Ty parasite extracts and the parental control after different durations of Rapa treatment. Membranes were probed with anti-Ty and anti-catalase (loading control) antibodies (*n* = 3). **c, d** Plaque assays of DiCre and iKD PBGD-Ty parasites in the normal medium or supplemented with 300 μM 5-aminolevulinic acid (5ALA) in the presence or absence of Rapa (**c**) and quantification of plaque sizes (*n* = 3) (**d**). **e** IFA of iKD IspH-Ty and iKD IspH-MVA-HA parasites stained with anti-HA, and anti-GAP45 antibodies. **f** Western blot of iKD

IspH-Ty and iKD IspH-MVA-HA parasite extracts, probed with anti-HA and anti-catalase (loading control) antibodies. **g, h** Plaque assays of iKD IspH-Ty and iKD IspH-MVA-HA parasites in the presence or absence of Rapa in normal medium (− MVL) or supplemented with 10 mM mevalonolactone (+ MVL) (**g**) and quantification of plaque sizes (*n* = 3) (**h**). Pictures in (**a–c**, **e–g**) are representative of three independent experiments. Bar graphs in (**d** and **h**) show the means (bars) of three independent experiments. Dots represent the means of the experiments, averaging at least 10 plaques per experiment. Error bars indicate the standard deviation and Student's two-sided *t* tests compare the indicated conditions. *p*-values are given and were considered significant at *p* < 0.05. Scale bars in a and e: 2 μm. Source data are provided as a Source Data file.

treatment (40 μM) resulted in the expected loss of lysis plaques (Fig. 4a, b), a severe growth defect (Fig. 4c), and dramatic loss of the apicoplast (Fig. 4d).

To potentially rescue actinonin-induced apicoplast loss, we engineered RH Δ*HXGPRT*Δ*KU80* parasites (RH) expressing the MVA cassette (RH-MVA-HA). Integration of the MVA cassette was confirmed by PCR (Supplementary Fig. 4a) and expression of the HA-tagged polyprotein was confirmed by IFA and western blot (Supplementary Fig. 4b, c). Of note, while the MVA cassette was observed close to the expected size of 113 kDa by western blot (Supplementary Fig. 4c), several bands corresponding to smaller proteins were also detected in RH-MVA-HA parasites, which were not observed in iKD IspH parasites expressing the cassette (Fig. 3f). These additional bands are likely the result of partial degradation. Importantly, however, most of the protein is detected at its full length. These RH-MVA-HA and RH parasites were treated with actinonin or not, in normal medium (NM, Dulbecco's Modified Eagle Medium, DMEM) or DMEM supplemented with FAs (FA C14:0 and C16:0, each 50 μM, coupled to BSA), LPA (0:0/14:0, 20 μM),

5ALA (300 μM), MVL (10 mM) and additional serum (7% final). The medium of this composition will be referred to as apicoplast-rescue medium (ARM), in the following. Remarkably, RH-MVA-HA, but not RH parasites, treated with actinonin formed clearly visible lysis plaques when cultured in ARM, but not NM (Fig. 4e, f), indicating a rescue of the apicoplast's functions. These plaques were, however, notably smaller compared to untreated controls (45% of plaque size compared to controls), consistent with the partial rescue observed when rescuing FA-, lipid- and heme synthesis (see Figs. 2c, d and 3c, d). To specifically identify the contribution of each metabolite to the bypass of the apicoplast, we withdrew individual or paired metabolites from the complete ARM (Supplementary Fig. 4d, e - note that this experiment shows the same +actinonin in ARM control as the experiment shown in main figure panel 4e). These results highlight that MVL and either FAs or LPA are critically needed to support the lytic cycle during apicoplast disruption. Withdrawing 5ALA led to the formation of considerably smaller plaques, suggesting that it also critically contributes to the organelle's bypass. Notably, removing either FAs or LPA had a modest

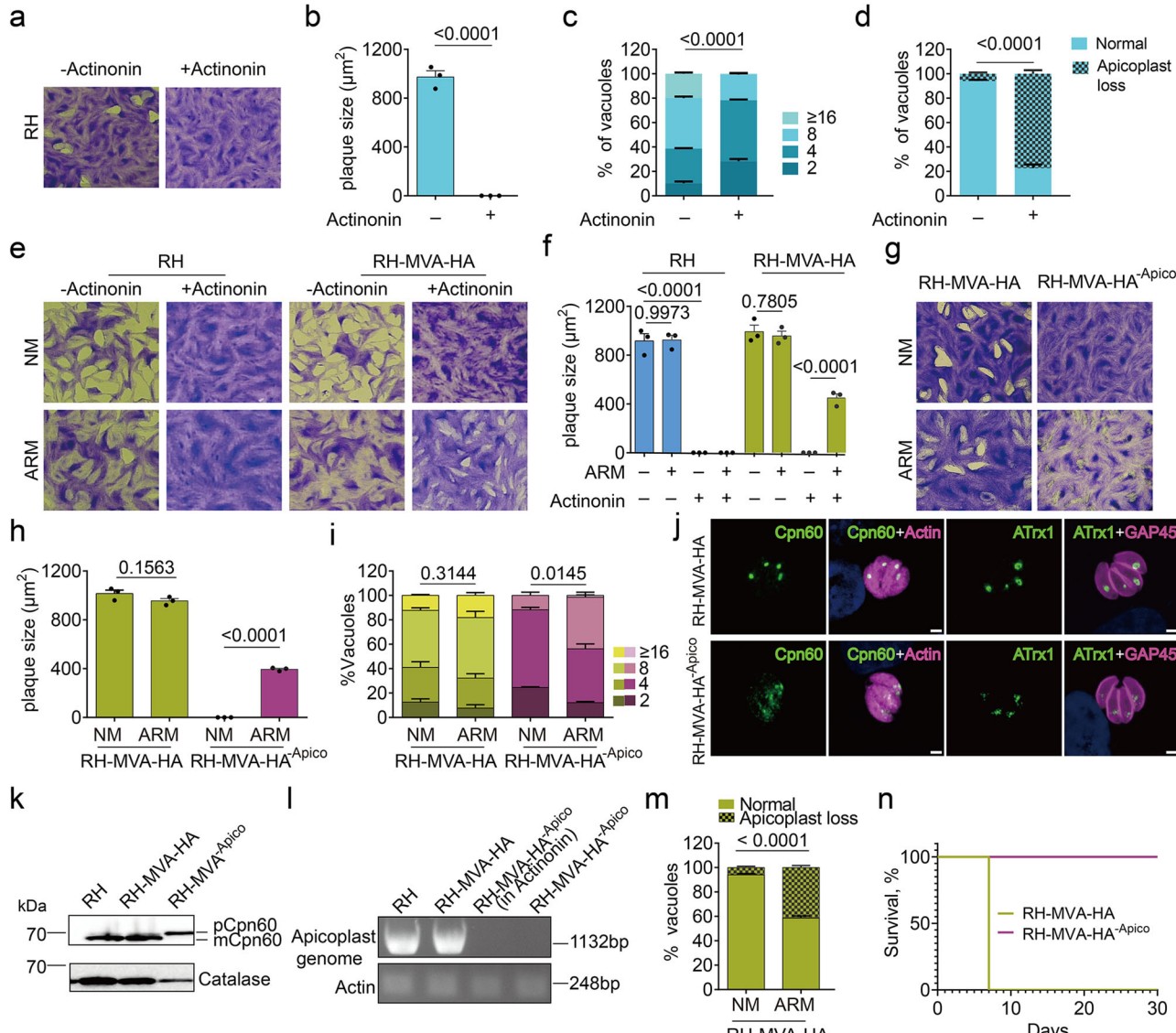

**Fig. 4 | Validation and characterization of *T. gondii* devoid of an apicoplast.**
**a**, **b** Plaque assay of RH in the absence or presence of 40 μM actinonin (**a**) and quantification of plaque sizes (*n* = 3) (**b**). **c** Intracellular growth of RH parasites, determined at 24 h after inoculation and a total duration of 72 h actinonin treatment (40 μM) or not (*n* = 3). **d** Quantification of apicoplast loss based on ATrx1 signal in RH following treatment as for c (*n* = 3). **e**, **f** Plaque assay of RH and RH-MVA-HA parasites in normal medium (NM) and apicoplast-rescue medium (ARM) ± actinonin (40 μM) (**e**) and quantification of plaque sizes (*n* = 3) (**f**). **g**, **h** Plaque assay of RH-MVA-HA and RH-MVA-HA⁻ᴬᵖⁱᶜᵒ parasites in NM and ARM (**g**) and quantification of plaque sizes (*n* = 3) (**h**). **i** Intracellular growth assay of RH-MVA-HA and RH-MVA-HA⁻ᴬᵖⁱᶜᵒ in NM or ARM (*n* = 3). **j** Indirect immunofluorescence assays (IFAs) of RH-MVA-HA and RH-MVA-HA⁻ᴬᵖⁱᶜᵒ parasites stained with anti-Cpn60 (apicoplast lumen marker) and anti-actin (cytosol marker) or anti-ATrx1 (apicoplast membrane marker) and anti-GAP45 (pellicle marker) antibodies. **k** Western blot of RH, RH-MVA-HA, and RH-MVA-HA⁻ᴬᵖⁱᶜᵒ parasite extracts.

Membranes were probed with anti-Cpn60 and anti-catalase (loading control) antibodies. The pre-processed (p) and mature (m) forms of Cpn60 are indicated. **l** PCR amplifying a fragment of the apicoplast genome and nuclear-encoded actin (control) in RH, RH-MVA-HA, and RH-MVA-HA⁻ᴬᵖⁱᶜᵒ parasites. The expected amplicon sizes are indicated. **m** Apicoplast loss in RH-MVA-HA following eight consecutive passages in NM or ARM as determined by ATrx1 signal (*n* = 3). **n** survival curve of mice infected with 1000 RH-MVA-HA or RH-MVA-HA⁻ᴬᵖⁱᶜᵒ parasites (3 mice per condition). Pictures in (**j**–**l**) are representative of three independent experiments. All bar graphs show the means (bars) of three or more independent experiments with error bars indicating the standard deviation. Dots in (**b**, **f** and **h**) represent the average of each experiment, averaging at least 10 plaques per experiment. Student's two-sided *t* tests compare the indicated conditions. *p*-values are given at were considered significant at p < 0.05. Scale bars in j: 2 μm. Source data are provided as a Source Data file.

but significant impact on the plaque sizes, suggesting that these are partially redundant and can potentially be interconverted outside the apicoplast. To provide the best possible growth conditions and avoid any long-term defects, we opted to supply the complete ARM during the following experiments.

Subsequently, we treated RH-MVA-HA parasites for multiple passages over 20 days with actinonin in ARM and purified clonal populations of parasites. These apicoplast-less parasites (RH-MVA-HA⁻ᴬᵖⁱᶜᵒ) could be maintained in ARM indefinitely (> 60 passages) and could be

frozen and thawed but are unable to survive in a normal medium (Fig. 4g, h). While RH-MVA-HA⁻ᴬᵖⁱᶜᵒ parasites form clearly visible plaques when cultured in ARM, their lysis plaques are smaller compared to those formed by control parasites possessing an apicoplast (41% of plaque size compared to controls) (Fig. 4g, h). Similarly, the replication rate of RH-MVA-HA⁻ᴬᵖⁱᶜᵒ parasites is reduced compared to parasites possessing the organelle but is markedly improved when grown in ARM (Fig. 4i). To validate that these clonal RH-MVA-HA⁻ᴬᵖⁱᶜᵒ have indeed lost their apicoplast, we carried out IFAs with the apicoplast

markers Cpn60 and ATrx1. Both markers show weaker and more diffuse staining above the Golgi, compared to parasites with an intact organelle (Fig. 4j). We further confirmed the absence of the organelle through electron microscopy. In a single section, encompassing two vacuoles of eight RH-MVA-HA parasites, at least seven apicoplasts were visible (Supplementary Fig. 5a). In contrast, no apicoplast was observed in a single section through two vacuoles encompassing eight RH-MVA-HA⁻ᴬᵖⁱᶜᵒ parasites (Supplementary Fig. 5a), as well as in a series of 16 consecutive sections of a single RH-MVA-HA⁻ᴬᵖⁱᶜᵒ parasite (Supplementary Fig. 5b). The absence of the organelle was further confirmed by western blot analysis, which revealed the presence of only the pre-processed form of Cpn60, with no detection of the mature protein (Fig. 4k). In addition, the loss of the organellar genome was confirmed by PCR amplification of a fragment of the 35 kb molecule, utilizing the primers specified in Supplementary Data 1 (Fig. 4l).

Following the successful rescue of the organelle's functions, we aimed to determine whether apicoplast loss only occurs as a consequence of drug treatment, by chemically disrupting the organelle, or if it could arise spontaneously under permissive conditions. To investigate this, we cultured RH-MVA-HA parasites in both normal medium and ARM for eight passages, in the absence of any drug. Remarkably, ~ 41% of vacuoles displayed either partial or complete apicoplast loss in ARM, in stark contrast to just 6% in normal medium (Fig. 4m). These findings indicate that apicoplast loss occurs frequently under permissive conditions, offering valuable insights into the evolutionary loss of this organelle, which has occurred multiple times across different lineages[60].

Lastly, to assess the virulence of apicoplast-less parasites, three mice were infected with 1000 parasites possessing the apicoplast (RH-MVA-HA) or not (RH-MVA-HA⁻ᴬᵖⁱᶜᵒ). The presence of viable parasites in the inoculum was validated by plaque assays (Supplementary Fig. 4f). Mice infected with RH-MVA-HA possessing an apicoplast presented severe symptoms seven days after the infection and had to be sacrificed, while the three mice infected with RH-MVA-HA⁻ᴬᵖⁱᶜᵒ presented no signs of infection and survived long-term (Fig. 4n). Subsequent analyses of the sera revealed that all but one mouse, infected with RH-MVA-HA⁻ᴬᵖⁱᶜᵒ, tested positive for seroconversion (Supplementary Fig. 4g). These results demonstrate that T. gondii lacking the apicoplast are avirulent, underscoring that apicoplast loss is irreversible and that not all bypass metabolites can be adequately salvaged under physiological conditions.

## Bypass of the apicoplast facilitates the identification of drugs targeting the organelle in T. gondii

Next, we assessed the sensitivity of T. gondii to different drugs under both standard and apicoplast bypass culture conditions, i.e., cultured in normal medium (NM) or apicoplast rescue medium (ARM), by performing plaque (Fig. 5a–f) and intracellular growth assays (Fig. 5g–i) in presence of clindamycin, atovaquone and triclosan. Under apicoplast-bypass conditions, RH-MVA-HA demonstrated significantly reduced sensitivity to clindamycin, a known inhibitor of apicoplast protein synthesis[61], as shown in plaque assays (Fig. 5a–d). Similarly, a modest but significant improvement upon apicoplast bypass conditions was observed in an intracellular growth assay (Fig. 5g), assessing growth over 24 h, and exposure to clindamycin (10 μM) for 36 h. Conversely, no difference was observed in the sensitivity to atovaquone, an inhibitor of the mitochondrial bc1 complex[62] by plaque (Fig. 5b–e) or intracellular growth assay (Fig. 5h). These findings confirm that the organelle's bypass in T. gondii can be utilized in drug screens to identify compounds targeting the organelle, similar to what has been done in P. falciparum[21–27,63]. Crucially, we propose that this bypass tool extends beyond what has been feasible working with blood-stage malaria parasites. Specifically, the bypass in T. gondii could potentially identify inhibitors of apicoplast pathways that are non-essential in

intraerythrocytic P. falciparum, such as FA and heme synthesis[29,30], but fitness conferring or essential in T. gondii[31,32,34].

To test this, we employed triclosan, a potent inhibitor of the FASII component FabI, which was believed to kill blood-stage malaria parasites by disrupting FA synthesis but was later found to act by exhibiting significant off-target effects[30]. In contrast to intraerythrocytic P. falciparum, T. gondii does rely on FA synthesis[31,32] and triclosan may primarily act by inhibiting this pathway. Indeed, RH-MVA-HA exhibited a reduced sensitivity to triclosan under apicoplast-bypass conditions, as observed in plaque (Fig. 5c, f) and intracellular growth assays (Fig. 5i), confirming the drug's impact on the organelle. These results establish the broad utility of this bypass, due to the organelle's extensive functions in T. gondii.

## Metabolomic analyses reveal absence of metabolic activity in remnant plastid vesicles of T. gondii

To gain a more comprehensive understanding of the metabolic consequences associated with apicoplast loss, we profiled the metabolome of RH, RH-MVA-HA, and RH-MVA-HA⁻ᴬᵖⁱᶜᵒ in normal medium (NM) and in apicoplast rescue medium (ARM), using GC-MS (Fig. 6a and Supplementary Data 3). It is important to note that RH-MVA-HA⁻ᴬᵖⁱᶜᵒ are not viable long-term in NM (see Fig. 4e–h) but could be sustained for the duration of this and the below described experiments (48 h). RH-MVA-HA⁻ᴬᵖⁱᶜᵒ, cultured in NM, showed numerous (17) significant changes in metabolite levels ($p < 0.05$, FC > 2) when compared to RH-MVA-HA parasites possessing the organelle (Fig. 6a). We observed a reduction in citric acid, which may be a direct consequence of the loss of the apicoplast given that the tricarboxylic acid (TCA) cycle enzyme aconitase has been localized to the organelle[64]. Instead, other changes, like an increase in myo-inositol, maybe a part of a general stress response[65]. Yet other changes are likely owed to the cells being previously cultured in distinct medium before transferring both strains to NM (e.g., increased 5ALA and MVL). Crucially, these changes were largely abrogated when comparing parasites with and without an apicoplast cultured in ARM. Comparing RH-MVA-HA and RH-MVA-HA⁻ᴬᵖⁱᶜᵒ in ARM, we observed significant changes in only two metabolites, namely increased levels of glutarate and MVL in parasites lacking the apicoplast (Fig. 6a). The increase in MVL is likely due to the extended culture period in ARM. These findings underscore the potential of apicoplast bypass medium to restore metabolic homeostasis in the absence of the organelle. Building on this metabolite profiling, we aimed to address a key question in the field: to what extent, if at all, does metabolic activity resume in the remnant vesicles following apicoplast loss? Previous studies have indicated that heme and CoA synthesis still occur in vesicles of P. falciparum, after disruption of the organelle[20,66].

First, we probed whether residual FA synthesis occurs in T. gondii lacking the organelle. To this end, intracellular parasites were labeled with ²H₂O-labeling followed by their purification and GC-MS analysis of parasite-derived FAs. The FA labeling rates of RH, RH-MVA-HA, and RH-MVA-HA⁻ᴬᵖⁱᶜᵒ were compared in NM and in ARM (Fig. 6b and Supplementary Data 4). Interestingly, the FA synthesis rate was markedly reduced in parasites when cultured in ARM, even in parasites that harbor the apicoplast (Fig. 6b). In RH-MVA-HA⁻ᴬᵖⁱᶜᵒ parasites, no labeling was detected in myristic acid (C14:0), but residual labeling was observed in palmitic acid (C16:0) (Fig. 6b), which likely reflects the uptake of FAs synthesized (labeled) by the host cells. However, residual FASII activity in the remnant vesicles or activity of the cytosolic FASI cannot be formally excluded using this approach.

To investigate this further, we labeled extracellular, purified T. gondii with U-¹³C₆-glucose (10 mM) for 5 h. Although FA synthesis is generally considered inactive in extracellular parasites[67], we detected low, but significant levels of fully ¹³C-labeled myristic acid isotopologues (M14) in parasites incubated with labeled glucose (3–7% isotopologue abundance after natural abundance correction) (Fig. 6c

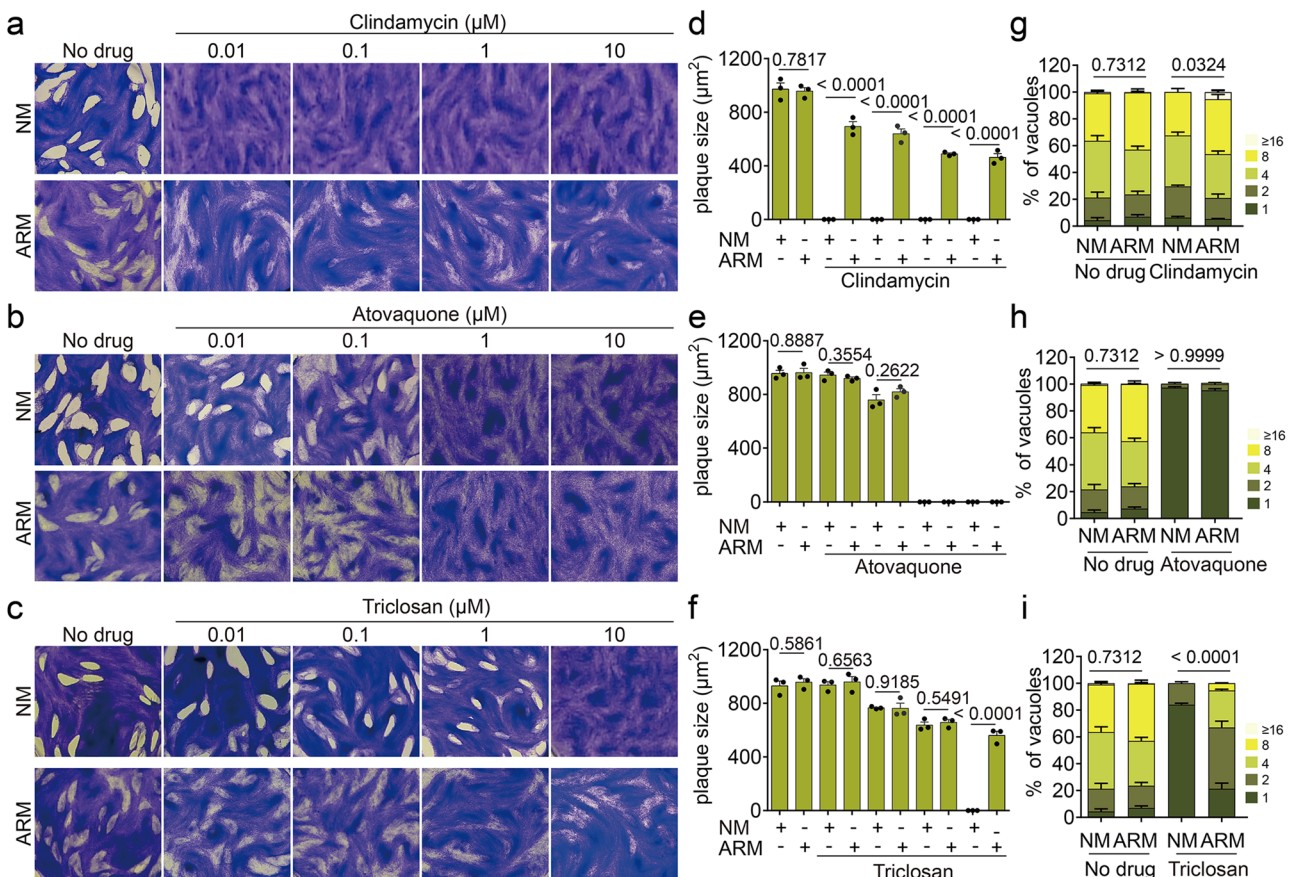

**Fig. 5 | *T. gondii* exhibit reduced sensitivity to apicoplast-targeting drugs under conditions bypassing the organelle. a–c** Plaque assays of RH-MVA-HA in the presence of different concentrations of clindamycin (**a**), atovaquone (**b**), and triclosan (**c**) cultured in normal medium (NM) or apicoplast rescue medium (ARM) and the corresponding plaque size quantifications ($n = 3$) (**d–f**). **g–i** Intracellular growth assays showing parasite numbers per vacuole following 24 h of growth in NM or ARM and a total of 36 h of drug exposure (10 μM) ($n = 3$). Note that the same results were used for the no-drug control in (**g–i**). Pictures in (**a–c**) are representative of three independent experiments. Bar graphs in (**d–f**) show the means (bars) of these three experiments, with dots averaging at least 10 plaques per experiment. Similarly, bar graphs in (**g–i**) show the means of three independent experiments. Error bars in (**d–i**) indicate the standard deviation, and Student's two-sided *t* tests compare the indicated conditions. In (**g–i**), the average number of parasites per vacuole was compared. *p*-values are given and were considered significant at $p < 0.05$. Source data are provided as a Source Data file.

and Supplementary Data 4). This labeling was present across different strains (RH and RH-MVA-HA) and culture media but was absent in parasites lacking an apicoplast (RH-MVA-HA$^{-Apico}$), strongly suggesting that FA synthesis is inactive following loss of the apicoplast.

Conversely, other pathways may continue to operate in the remnant vesicles. This is especially plausible for heme synthesis, as the enzymes involved, unlike those functioning in IPP and FA synthesis (IspG, IspH, and LipA), do not rely on iron-sulfur clusters [Fe-S]. The [Fe-S]-dependent enzymes in the apicoplast are presumably non-functional after the organelle is lost since SufB, a key component of the [Fe-S] synthesis pathway, is encoded by the plastid genome[68]. To investigate potential heme synthesis activity in apicoplast-less parasites, we generated mutants lacking the heme synthesis enzyme uroporphyrinogen III synthase (UROS, RH-MVA-HAΔ*UROS*) using the strategy shown in Supplementary Fig 6a, with primers listed in Supplementary Data 1. To ensure the recovery of Δ*UROS* parasites, despite the fact that heme synthesis is highly fitness-conferring in *T. gondii*[31,56] and that the lack of UROS can result in the non-enzymatic formation of toxic uroporphyrinogen I[69], the transfected cells were selected and cloned in ARM. The deletion of the *UROS* gene was validated by genomic PCR (Supplementary Fig 6b), using primers listed in Supplementary Data 1. Remarkably, the recovered clonal parasite lines had, without exposure to an apicoplast-disrupting drug, lost the apicoplast, consistent with the spontaneous apicoplast loss under permissive conditions described above. Thus, the strain obtained was termed RH-MVA-HA$^{-Apico}$Δ*UROS*. We then compared the growth rates of parasites lacking only the apicoplast with those lacking both, the apicoplast and *UROS*, and found no difference (Fig. 6d). While these data do not exclude the possibility of heme synthesis occurring in the remnant vesicles, they indicate that heme synthesis activity does not contribute to the fitness of parasites without an apicoplast. Instead, RH-MVA-HA$^{-Apico}$ parasites seem to rely exclusively on the uptake of heme and/or its late-stage precursors from the host.

## Apicoplast proteins display reduced abundance in parasites devoid of the organelle

To identify potential changes in the proteome of parasites missing the apicoplast, comparative proteomic analyses were performed with RH, RH-MVA-HA, and RH-MVA-HA$^{-Apico}$ parasites grown in normal medium (NM) or in apicoplast rescue medium (ARM). In each analysis, more than 4000 proteins were detected, and their relative abundances compared (Supplementary Data 5).

Remarkably, proteomic analyses revealed significant differences ($p < 0.05$, FC > 1.5) in the abundance of 828 proteins, when RH-MVA-HA parasites were cultured in ARM, compared to NM (Supplementary Data 5). Most of these *medium-dependent proteins* were upregulated in parasites cultured in ARM. Inspecting these affected proteins closer, we found that they largely localize to the ER (Supplementary Data 5)

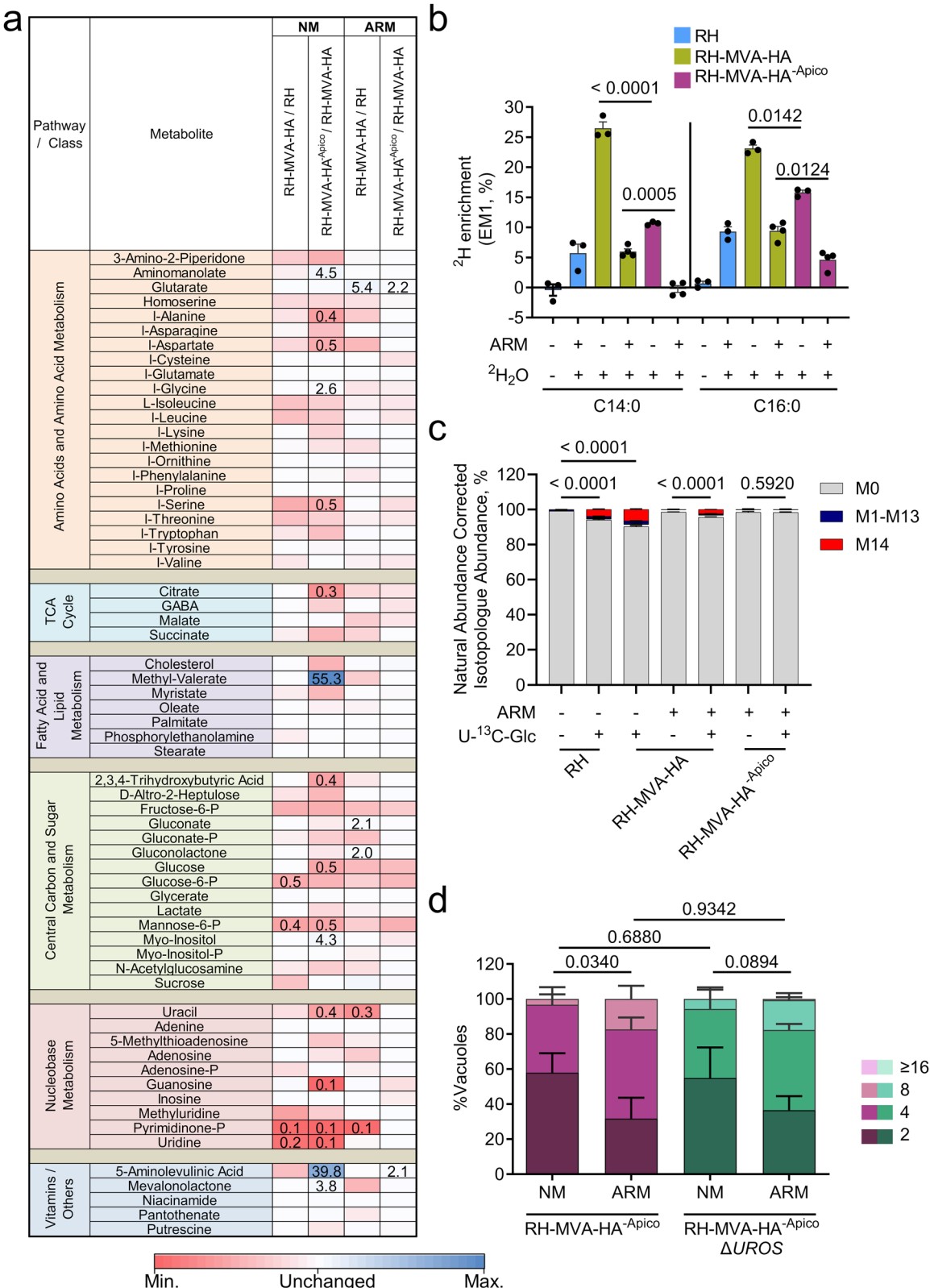

based on the LOPIT dataset on ToxoDB.org[70,71] and approximately 20% contain at least one transmembrane domain (TMD) (Supplementary Data 5).

In contrast, only a relatively small number of proteins (48) showed significant changes in their abundance in RH-MVA-HA parasites, compared to RH parasites. Most of these *strain-dependent*

*proteins* were upregulated in RH-MVA-HA parasites (Supplementary Data 5).

A comparison of parasites with (RH-MVA-HA) and without an apicoplast (RH-MVA-HA$^{-Apico}$), both cultured in ARM, revealed 244 proteins significantly altered in their abundance ($p < 0.05$, FC > 1.5) in parasites lacking the organelle (Fig. 7a). Most of these

**Fig. 6 | Metabolomic analyses of *T. gondii* lacking an apicoplast. a** Metabolomic profiling of RH, RH-MVA-HA, and RH-MVA-HA⁻ᴬᵖⁱᶜᵒ in normal medium (NM) and apicoplast rescue medium (ARM). Metabolites measured by GC-MS are categorized into distinct classes, with relative abundances displayed in a heatmap. Metabolite abundances were compared between the indicated conditions and tested for significant differences using Student's two-sided *t* tests. Significantly altered metabolites ($p < 0.05$, FC > 2) are highlighted by the provision of the numerical fold change in the corresponding cell ($n = 5$). **b** $^2H_2O$-labeling of intracellular RH, RH-MVA-HA, and RH-MVA-HA⁻ᴬᵖⁱᶜᵒ parasites over 24 h, followed by fatty acid (myristic acid, C14:0, and palmitic acid, C16:0) labeling analysis (excess molar enrichment of the M1 isotopologue, EM1 in %) via GC-MS ($n = 3 - 4$). **c** U–$^{13}C_6$-glucose labeling of purified extracellular parasites for 5 h, with isotopologue abundances measured by GC-MS ($n = 3$). **d** Intracellular growth assay comparing the growth rates of RH-MVA⁻ᴬᵖⁱᶜᵒ and RH-MVA-HA⁻ᴬᵖⁱᶜᵒΔ*UROS* parasites in NM and ARM ($n = 3$). Bar graphs in (**b**–**d**) represent the means of three or more independent experiments. Dots in b represent the $^2H$-enrichment calculated for each of the 3 – 4 independent biological replicates. All error bars denote the standard deviation. Student's two-sided *t* tests were used to compare conditions, with *p*-values < 0.05 considered significant. Source data are provided as a Source Data file.

*apicoplast-dependent proteins* were downregulated (211 proteins, 86%) (Fig. 7a). Using the LOPIT dataset to interrogate the subcellular localization of these altered proteins[71], many proteins (80) proved to be nuclear encoded putative apicoplast proteins, which were exclusively downregulated (Fig. 7b). These include many bona fide apicoplast proteins (e.g., Cpn60) but also proteins that have only recently been predicted by LOPIT to be at the apicoplast (e.g., hypothetical protein TGGT1_264200)[71]. The 80 downregulated apicoplast proteins present a considerable portion (62%) of the 129 proteins identified as tentative apicoplast proteins based on LOPIT. Notably, no apicoplast genome-encoded proteins were detected in any of the analyzed conditions, highlighting their low level of expression.

To evaluate the extent and specificity of the downregulation of nuclear-encoded apicoplast proteins in RH-MVA-HA⁻ᴬᵖⁱᶜᵒ compared to RH-MVA-HA parasites, we specifically assessed the relative abundance of the metabolic enzymes inside the apicoplast as well as of enzymes that co-operate with the apicoplast's metabolic pathways but localize to the mitochondrion, cytosol, or ER. Remarkably, out of the 33 metabolic enzymes within the apicoplast detected in our proteomics approach, most proteins (27, 82%) were downregulated ($p < 0.05$, FC > 1.5) (Fig. 7c and Supplementary Data 5). Only six proteins remained unchanged in abundance, two of which, related to the [Fe-S] synthesis pathway, may in fact not localize to the apicoplast[44]. Several (13) metabolic enzymes that operate outside the apicoplast but cooperate with its pathways were unchanged in their abundance (12 proteins), or, in one instance, upregulated (Fig. 7c). Collectively, these findings suggest that loss of the apicoplast leads to a significant and relatively specific decrease in many nuclear-encoded apicoplast proteins.

The downregulation of various nuclear-encoded apicoplast proteins in parasites lacking the organelle suggests that these proteins may be expressed at lower levels or that their stability is compromised due to improper trafficking. To distinguish between these two possibilities, we performed transcriptomic analyses comparing RH-MVA-HA and RH-MVA-HA⁻ᴬᵖⁱᶜᵒ parasites. This analysis revealed 753 differentially expressed transcripts ($p < 0.05$, FC > 2), with 596 transcripts downregulated in RH-MVA-HA⁻ᴬᵖⁱᶜᵒ parasites (Fig. 7d, Supplementary Data 6). Notably, 671 of the altered transcripts lacked assigned subcellular localization according to LOPIT (Supplementary Data 6)[71]. Gene Ontology (GO) term enrichment analysis indicated that alpha tubulin and cyclic nucleotide biosynthetic processes were the most significantly affected, based on only one and three impacted transcripts, respectively (Supplementary Data 6). This limited insight into the localization of the encoded proteins, along with the absence of clearly defined affected biological processes, complicates the interpretation of these transcriptional changes. Similar to the proteomic analysis, apicoplast genome-encoded transcripts were detected at very low levels across all strains and were therefore excluded from this analysis. Importantly, amongst the many altered transcripts, we identified only a single nuclear-encoded transcript of an apicoplast-localized protein.

Relatedly, when plotting the fold changes of proteins versus RNA transcripts from the comparison of RH-MVA-HA and RH-MVA-HA⁻ᴬᵖⁱᶜᵒ parasites, we observed a weak correlation between the two datasets (Fig. 7e), with only 20 genes showing changes at both RNA and protein levels. These findings reveal that the decreased abundance of apicoplast proteins in absence of the organelle is not due to their reduced transcription but the result of impaired protein stability.

## Proteomic analysis of apicoplast-less parasites facilitates the identification of novel essential apicoplast proteins

We hypothesized that the up- or downregulation of non-apicoplast proteins in response to the organelle's loss may provide important insights into coping mechanisms, e.g., the enhanced uptake of bypass metabolites. Additionally, the list of significantly downregulated proteins, which were either not assigned to a subcellular localization by LOPIT[71] (34 proteins) or which were localized to compartments other than the apicoplast (97 proteins), may include many unidentified apicoplast proteins. Indeed, literature searches revealed additional apicoplast proteins amongst the significantly downregulated proteins in apicoplast-less parasites that were not localized to the organelle by LOPIT. These include the leucyl-tRNA synthetase (LeuRS2) and a major facilitator superfamily (MFS) transporter that have been localized to the apicoplast by epitope tagging[72,73]. Hence, we speculated that this dataset may facilitate the discovery of unknown apicoplast proteins.

To test this, we selected a bona fide apicoplast protein (FtsH1), a protein that was tentatively localized to the apicoplast by LOPIT[71] (TGGT1_201270) and a protein of unknown subcellular localization (TGGT1_248770) from the list of significantly downregulated proteins in *T. gondii*⁻ᴬᵖⁱᶜᵒ. The known apicoplast protein and target of actinonin, FtsH1[58,71,74], which was reduced 1.57-fold in our proteomic analysis of apicoplast-less parasites, served as a reference here. TGGT1_201270 and TGGT1_248770 were selected based on their classification as hypothetical proteins with unknown functions[70], their potential essentiality (CRISPR/Cas9 fitness scores below −3)[48] as well as their marked downregulation following apicoplast loss (4.37 and 9.33-fold, respectively) (Supplementary Data 5).

To obtain insights into these proteins' localization and role for parasite fitness, each locus was modified as described above, generating iKD FtsH1-Ty, iKD 201270-Ty, and iKD 248770-Ty parasites. All proteins, including the candidate of unknown localization (TGGT1_248770), were found to localize to the apicoplast by IFA (Fig. 8a), confirming that the above proteomic analysis can facilitate the discovery of unknown apicoplast proteins based on their reduced abundance in parasites devoid of the organelle. We further characterized the importance of the three selected proteins. Their efficient downregulation after 48–72 h of rapamycin treatment was validated by IFA and western blot (Fig. 8b, c) with the pre-processed (p) and the mature (m) epitope-tagged proteins running close to, or slightly below, their predicted molecular weights (mFtsH1 - 143 kDa; p201270 - 110 kDa, m201270 - 65 kDa kDa; p248770 - 78 kDa, m248770 - 50 kDa) (Fig. 8c). The downregulation of each protein severely disrupted the parasite's lytic cycle, as evidenced by the absence of visible plaques of lysis (Fig. 8d, e).

Since this is the first report of TGGT1_248770 localizing to the apicoplast and identifying its essential role for the lytic cycle of *T. gondii*, we characterized this protein further. Based on the information

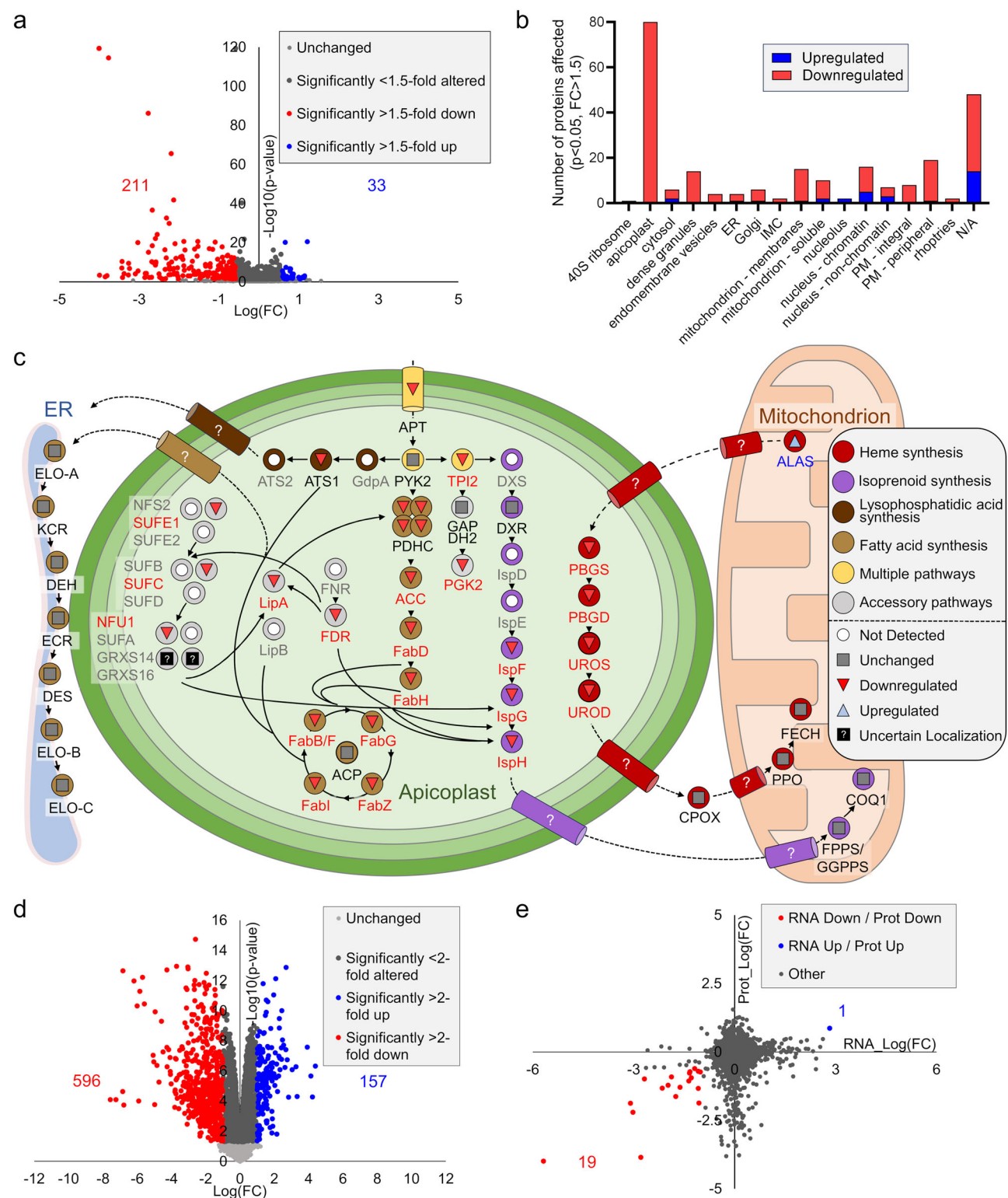

available on ToxoDB, TGGT1_248770 appears to be constitutively expressed and has no predicted domains[70]. Notably, TGGT1_248770 is only conserved in Coccidia, with homologs found in *Neospora*, *Besnoitia*, *Cystoisospora,* and *Cardiosporidium*[75] but absent in other apicomplexans. Structural searches via FoldSeek[76] based on Alphafold[77] predictions did not reveal any structural homologs in other phyla. An intracellular growth assay confirmed a critical role of TGGT1_248770 for parasites replication (Fig. 8f), comparable to that of FtsH1 (Supplementary Fig 7a). While TGGT1_248770 is not predicted to have any

TMDs by the prediction tool integrated on ToxoDB[70], other tools predict the presence of one or more TMDs near its C-term[78,79]. Indeed, fractionation experiments revealed its behavior as an integral membrane protein, soluble only in presence of detergent such as Triton-X100 (Fig. 8g). Furthermore, we found that the downregulation of TGGT1_248770 led to significant apicoplast loss, which was quantified based on the staining with anti-ATrx1 and anti-Cpn60 antibodies as well as DAPI, further corroborating the protein's role within the organelle (Fig. 8h). The apicoplast loss associated with TGGT1_248770

**Fig. 7 | Loss of the apicoplast leads to a reduction in the level of its constituents, likely due to reduced protein stability. a** Volcano plot highlighting changes in protein levels of parasites lacking an apicoplast (RH-MVA-HA⁻ᴬᵖⁱᶜᵒ) in comparison to their control with an apicoplast (RH-MVA-HA), both cultured in apicoplast rescue medium (ARM). Differential abundance testing between the two conditions was performed using an unpaired *t* test. Non-significantly or <1.5-fold altered proteins are displayed in gray, while significantly decreased or increased proteins (*p*-value < 0.05, FC > 1.5) are displayed in red and blue, respectively (*n* = 3). **b** Significantly increased or decreased proteins sorted according to their subcellular localization using the hyperplexed Localization of Organelle Proteins by Isotopic Tagging (LOPIT) data available under https://proteome.shinyapps.io/toxolopittzex/. **c** Overview of proteins associated with metabolic pathways in the apicoplast and related enzymes in the cytosol, mitochondrion, and at the endoplasmic reticulum (ER). Symbols and colors (see legend) indicate each protein's behavior in the comparative proteomic analysis plotted in (**a**). **d** Volcano plot highlighting changes in transcript levels of parasites lacking an apicoplast (RH-MVA-HA⁻ᴬᵖⁱᶜᵒ) in comparison to their control strain with an apicoplast (RH-MVA-HA), both cultured in apicoplast rescue medium (ARM) (*n* = 4). A differential expression analysis was performed with the statistical analysis R/Bioconductor package edgeR1.38.4. with multiple testing Benjamini and Hochberg correction FDR 5% and a fold change threshold of 2. Non-significantly or <2-fold altered transcripts are displayed in gray, while significantly decreased or increased transcripts (*p*-value < 0.05, FC > 2) are displayed in blue and red, respectively. **e** Log fold changes from proteomic analyses plotted against log fold changes in transcriptomic analyses from the comparison of RH-MVA-HA and RH-MVA-HA⁻ᴬᵖⁱᶜᵒ parasites cultured in ARM (as plotted in **a** and **d**). Genes affected significantly at the RNA and protein level (*p* < 0.05, FC > 2 for transcriptomics, and FC > 1.5 for proteomics) are highlighted in red or blue for down- and upregulated products, respectively. Note that for the assessment of the subcellular localization (**b**), some compartments reported as distinct in the original study were merged for simplicity: apical1/apical2, rhoptries1/rhoptries2, PM – peripheral1/ PM – peripheral 2 and ER1/ER2. For abbreviations related to the enzymes in panel (**c**), please refer to the Supplementary Information file.

downregulation was, however, less severe than observed for FtsH1 (Supplementary Fig 7b). Given the lack of identifiable domains, sequence and structural homologs, the essential function of this coccidian apicoplast protein remains unknown and must be elucidated in future studies.

## Discussion

Ever since its discovery[14], the apicoplast has garnered considerable attention due to its appeal as a drug target in many disease-causing Apicomplexa, including *P. falciparum* and *T. gondii*[15]. While the acquisition of plastids has occurred several times independently throughout evolution, loss of the organelle is difficult to prove and a rare event[60,80]. Plastid loss has only been definitively demonstrated in two apicomplexans (*Cryptosporidium* and some gregarines) as well as in the dinoflagellate *Hematodinium*[80]. Importantly, plastids are not just acquired or lost, but encounter diverse fates, resulting in plastids with varying functions and possessing a genome or not[60,80–82]. In arguably the best studied apicomplexans, *T. gondii* and *P. falciparum*, the functions performed by the apicoplast are similar. However, the importance of each pathway is highly dependent on the life cycle stage and niche of the parasite. This is highlighted by differences in the apicoplast's functions between the *P. falciparum* intraerythrocytic stage, which only relies on IPP synthesis[18], and its liver and mosquito stages, which additionally rely on FA and heme synthesis[18,29,30,83].

The fast-replicating *T. gondii* tachyzoites, responsible for acute infection, are expected to rely on the four output-producing apicoplast pathways. We hypothesized that these could be bypassed directly or indirectly in tissue culture based on previous reports and set out to investigate this systematically here. FA and lipid precursor synthesis had already been shown to be partially rescued through supplementation of their products[31,33,84], which was validated here. Previously, we have demonstrated that the function of the cytosolic heme synthesis enzyme coproporphyrinogen III oxidase (CPOX) can be bypassed by supplementing the first heme precursor, 5ALA. This likely occurs through upregulation of heme synthesis in the host, facilitating the salvage of heme intermediates by the parasite[31]. Similarly, intraerythrocytic *P. falciparum* have also been shown to salvage and metabolize heme synthesis intermediates from their host, effectively bypassing the first six steps of synthesis within the parasite[85]. However, it remained unclear whether a similar bypass would be possible upon disruption of an apicoplast-resident heme synthesis enzyme or if potentially toxic precursors accumulate, causing parasite death. Here, we provide evidence that disruption of the apicoplast-resident PBGD can be equally bypassed by 5ALA supplementation, consistent with our previous hypothesis that only *Tg*FECH is essential[31,34]. A previous investigation of the heme synthesis enzyme, uroporphyrinogen III decarboxylase (UROD) revealed

the importance of the apicoplast-localizing portion of this pathway, but, in fact, the data by Tjihn et al. showed continued, albeit slow growth of the parasites following depletion of the enzyme[35]. This further suggests that, although heme synthesis is highly fitness-conferring, *T. gondii* can partially meet its heme requirements by salvage.

We also report here a complete rescue of the IPP synthesis pathway. This was achieved through expression of a mevalonate bypass cassette[28,45], which enables IPP synthesis independent of the endogenous MEP/DOXP pathway. This cassette has previously been employed, but the effectiveness of this rescue remained unclear, given that the bypassed enzyme, the triose phosphate isomerase 2, is not directly involved in IPP synthesis and considering the fact that the rescue was only partial (plaque sizes <50% compared to controls)[45]. Notably, rescuing this pathway in *T. gondii* requires the provision of 10 mM mevalonolactone (MVL). In comparison, 1,000-fold lower levels of MVL (10 μM) were sufficient to support growth of *P. falciparum* expressing the MVA cassette, when inhibited in their endogenous IPP synthesis pathway[28]. This discrepancy highlights differences in host cell or parasite plasma membrane permeability and/or differences in expression or effectiveness of the complemented protein. It is likely that the efficient uptake of MVL by *P. falciparum* is facilitated by the parasite-induced new permeation pathway, which increases the red blood cell permeability[86].

The importance of IPP synthesis in *T. gondii*, although well-documented[36,38], still raises several questions. In *P. falciparum*, it was demonstrated that isoprenoids, specifically very long chain isoprenoids, are needed for apicoplast biogenesis[23]. We demonstrate here that this is not the case in *T. gondii*. Loss of IspH is detrimental for *T. gondii* but does not lead to apicoplast loss, within the time frame tested here. Consistent with this, the closest homolog of the enzyme responsible for generating long chain isoprenoids in the *P. falciparum* apicoplast[23] does not localize to the apicoplast in *T. gondii*, but rather to the mitochondrion, where it has been shown to be needed for mitochondrial respiration[42]. Our proteomic analyses further corroborate these results, by showing no decrease, but in fact a slight increase in the abundance of the heptaprenyl diphosphate synthase (Coq1) in parasites lacking the apicoplast. Furthermore, depletion of MiaA, which is required for tRNA prenylation in the apicoplast, was shown to be inconsequential for the apicoplast in *P. falciparum*[23] and similar results can be expected for *T. gondii*, given its fitness score of −0.48[48], suggesting a dispensable function. Additionally, *T. gondii* was previously shown to salvage longer chain isoprenoids, such as farnesyl diphosphate (FPP) and geranylgeranyl diphosphate (GGPP), efficiently from its host, tolerating disruption of their synthesis[84]. Thus, several key functions of isoprenoids are either not essential or can be bypassed by salvage from the host. Together, our results, and previous

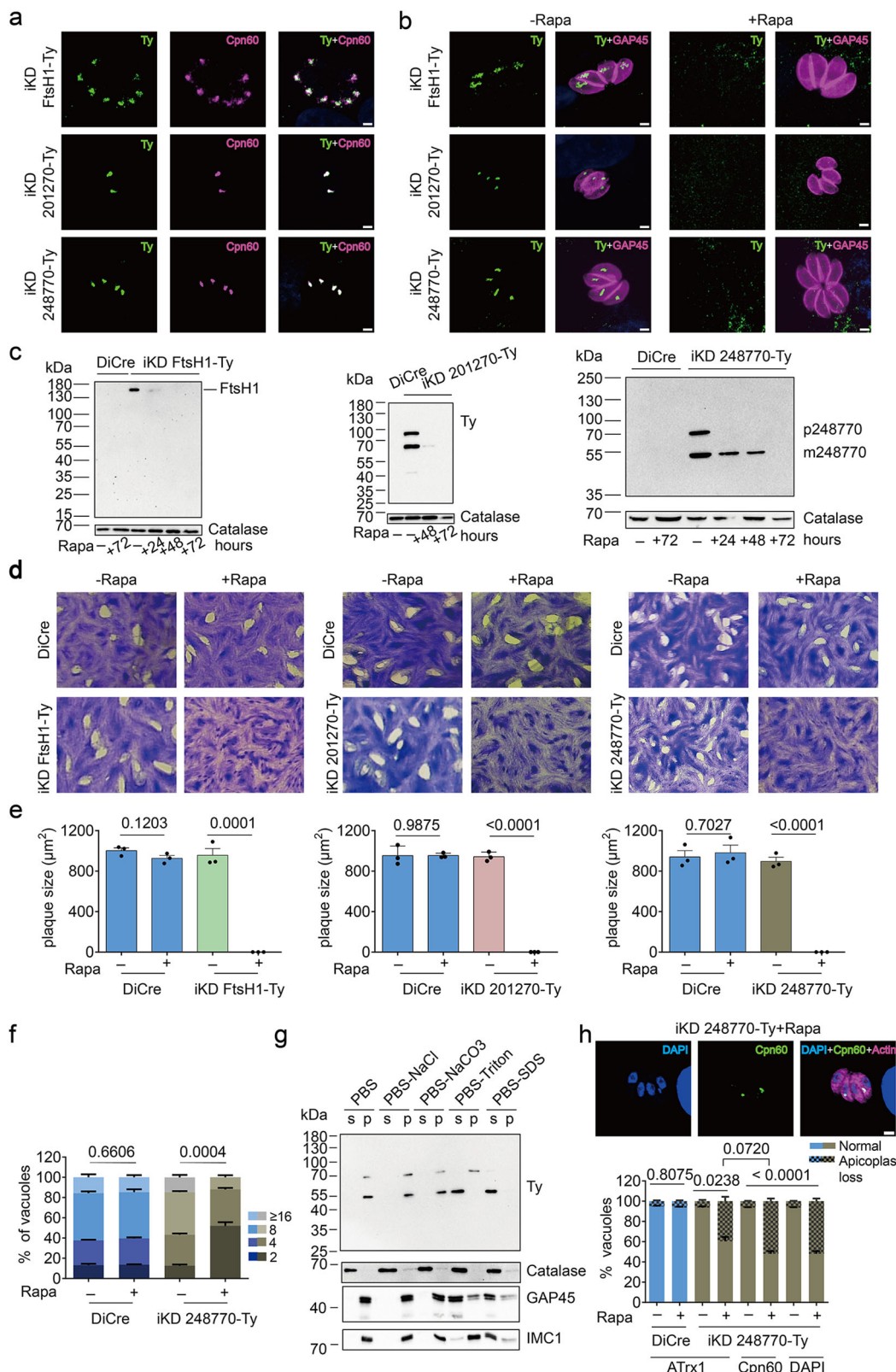

findings indicate that isoprenoids are not required for the *T. gondii* apicoplast and IPP may primarily serve to support the synthesis of ubiquinone[42]. Additional essential roles of endogenously produced IPP/DMAPP could be ruled out, by disrupting the MEP/DOXP synthesis pathway within the apicoplast and attempting its rescue by provision with exogenous long chain ubiquinone[42].

Upon bypassing all four output-generating metabolic pathways in the apicoplast, an incomplete but significant rescue of *T. gondii* fitness and growth was achieved. These findings uncover that the *T. gondii* apicoplast does not harbor additional, essential pathways that generate vital metabolites for *T. gondii*. Whether the incompleteness of the observed rescue is a matter of dosage of the supplemented

**Fig. 8 | Characterization of a newly identified Coccidia-specific apicoplast protein. a** Indirect immunofluorescence assays (IFAs) of iKD FtsH1-Ty, iKD 201270-Ty, and iKD 248770-Ty parasites, stained with anti-Ty and anti-Cpn60 (apicoplast lumen marker) antibodies. **b** IFAs of iKD FtsH1-Ty, iKD 201270-Ty, and iKD 248770-Ty parasites treated with rapamycin (+ Rapa, 72 h) or not and stained with anti-Ty and anti-GAP45 (pellicle marker) antibodies. **c** Western blots of DiCre, iKD FtsH1-Ty, iKD 201270-Ty, and iKD 248770-Ty parasite extracts after different durations of Rapa treatment. Membranes were probed with anti-Ty and anti-catalase (loading control) antibodies ($n = 3$). **d, e** Plaque assays of DiCre, iKD FtsH1-Ty, iKD 201270-Ty and iKD 248770-Ty parasite in culture medium ± Rapa (**d**) and quantification of plaque sizes ($n = 3$). **e, f** Intracellular growth assay of DiCre and iKD 248770-Ty parasites. Data reflects intracellular growth over 24 h and the total duration of Rapa treatment of 72 h ($n = 3$). **g** Solubility assay of extracts from iKD 248770-Ty

parasites. Membranes were probed with anti-Ty antibodies. Additionally, membranes were probed with anti-catalase, anti-GAP45, and anti-IMC1 antibodies as markers for the PBS-, Triton-, and SDS-soluble fraction, respectively. s, supernatant; p, pellet. **h** Quantification of apicoplast loss based on ATrx1 (apicoplast membrane marker), Cpn60, and DAPI staining in DiCre and iKD 248770-Ty parasites, following treatment as in f ($n = 3$). Pictures in (**a**, **b**, and **g**) are representative of three independent experiments. All bar graphs show the means (bars) of three independent experiments. In e, dots represent the means of each experiment, averaging at least 10 plaques per experiment. Error bars indicate the standard deviation. Student's two-sided $t$ tests compare the indicated conditions. $p$-values are given and were considered significant at $p < 0.05$. Scale bars in (**a**, **b** and **h**): 2 μm. Source data are provided as a Source Data file.

metabolites, or if it is a consequence of the lack of other fitness-conferring metabolites or the effect of general stress following loss of the organelle remains unclear. Regardless, we show that the reduced fitness of *T. gondii*[−Apico] parasites does not limit their applicability in various functional analyses, including drug assays, genetic studies and a range of omics approaches. However, the reduced fitness of *T. gondii*[−Apico] may restrict their suitability in comparative screenings that rely on equal growth rates, such as genome-wide CRISPR/Cas9 fitness screens. This limitation could potentially be overcome by optimizing the composition of the bypass media to enhance its effectiveness and improve the fitness of *T. gondii*[−Apico], broadening the potential applications of this model.

As seen in *P. falciparum*[18], the loss of the apicoplast in *T. gondii* results in the formation of remnant vesicles in place of the intact organelle. The structure and function of these vesicles remain unclear. In *P. falciparum*, it has been speculated that these vesicles may retain some metabolic activity[20,66]. However, given that the apicoplast genome encodes SufB, a critical component of [Fe-S] synthesis, the loss of the plastid and its genome inevitably results in a loss of the organellar [Fe-S] synthesis pathway and the enzymes dependent on it such as LipA, IspG and IspH[68]. To investigate potential FA or heme synthesis activity in the vesicles, we employed $^2$H and $^{13}$C stable isotope labeling followed by mass spectrometry as well as genetic techniques, deleting the heme synthesis enzyme uroporphyrinogen III synthase (UROS). Our findings indicate that FA synthesis does not occur at detectable levels, and any residual heme synthesis does not contribute to the fitness of the parasite following the loss of the apicoplast. These results provide clear evidence that apicoplast-associated metabolic activity largely stalls upon loss of the organelle in *T. gondii*. A more detailed investigation concerning the biogenesis, structure, and function of these remnant vesicles is warranted.

In further characterizing *T. gondii* devoid of the apicoplast, we provide the first evidence that its loss leads to a significant reduction in numerous nuclear-encoded proteins typically localized there. Notably, this reduction is not accompanied by a decrease in the levels of nuclear transcripts encoding apicoplast proteins, equivalent to what has been reported for *P. falciparum* lacking the organelle[28]. Thus, the decline in apicoplast-resident proteins is likely owed to their instability resulting from failed trafficking. We present a table listing 131 proteins that are downregulated in parasites lacking the organelle, which have not been previously associated with the apicoplast and may largely represent unidentified apicoplast proteins. This list serves as a valuable resource for identifying novel apicoplast proteins, as exemplified by our investigation of a candidate protein found to be an essential apicoplast protein in Coccidia.

Besides aiding in the discovery of new apicoplast proteins, the proteomic analysis of apicoplast-less *T. gondii* also provides insights into the interconnection between the two organelles of endosymbiotic origin, found in most apicomplexans: Among the proteins that were downregulated upon apicoplast loss were several mitochondrial proteins, specifically eight and four components of

complex III and IV of the respiratory chain, respectively (highlighted in red in Supplementary Data 5)[87,88]. Strikingly, these were not found to be altered significantly at the transcript level (Supplementary Data 6), indicating changes occurring at the translational or protein stability level in the absence of the apicoplast. The downregulation of several components of these complexes is expected to result in a reduction in mitochondrial respiration[89], although not specifically tested here.

We speculate that the drop in complex III and IV components is likely owed to a lack of heme. While supplementation of 5ALA alleviates some of the fitness defects observed for parasites deficient in heme synthesis, this rescue is incomplete, suggesting a persisting lack of heme in parasites depleted in a specific heme synthesis enzyme or lacking the apicoplast entirely. In Apicomplexa, complexes III and IV are the only heme-bearing complexes of the mitochondrial respiratory chain[90] and lack of heme has been shown to destabilize these complexes in mammalian cells, particularly complex IV[91,92]. Intraerythrocytic *P. falciparum* parasites, which in contrast to *T. gondii* do not rely on heme synthesis[93], exhibit full fitness when losing the apicoplast in the presence of IPP or mevalonolactone[18,28]. In *T. gondii*, we demonstrate that mevalonolactone supplementation in MVA cassette-expressing parasites fully restores the lytic cycle defect associated with a defect in the endogenous IPP synthesis pathway. Hence, an isoprenoid-related defect such as a lack in ubiquinone[42] appears unlikely to account for the drop in mitochondrial respiratory chain components in *T.gondii*[−Apico] but cannot be excluded. Notably, the assembly and stability of complex III and IV, but not complex II and V, relies on proteins encoded on the minimal mitochondrial genome of *T. gondii*[94]. Therefore, a decrease in the abundance of several components of complex III and IV, as observed here, also occurs upon defects in mitochondrial translation[89]. Hence, an unknown impact of the apicoplast on mitochondrial translation cannot be excluded.

Furthermore, changes at the proteome level upon apicoplast loss also offer valuable insights into the adaptive mechanisms that enable parasite survival in the absence of this essential metabolic hub. Similar adaptations likely occurred concomitant with plastid loss in the ancestor of *Cryptosporidium* spp.[60]. These responses may be reflected in the upregulation or downregulation of proteins that function outside the apicoplast, such as the upregulation of plasma membrane transporters necessary for the uptake of metabolites that compensate for the loss of apicoplast functions. However, among the upregulated proteins in apicoplast-less parasites, were only two proteins harboring a single TMD and no proteins containing multiple TMDs. In contrast, several putative transporters with multiple TMDs at the plasma membrane, Golgi, and endoplasmic reticulum were upregulated when parasites were cultured in the apicoplast rescue medium (ARM). These findings suggest that adaptations outside the apicoplast are primarily shaped by nutrient availability.

Interestingly, despite these extensive adaptations outside of the organelle, in parasites possessing an apicoplast, only seven apicoplast proteins were downregulated in response to exposure to apicoplast

rescue medium, while 14 were upregulated (see Proteomic changes due to distinct media, RH-MVA-HA cultured in normal medium vs ARM, Supplementary Data 5). This suggests that *T. gondii* continues to rely on the organelle and its pathways, even when alternative metabolites are available, consistent with the notion that parasites devoid of the organelle exhibit reduced fitness even under conditions bypassing the organelle. Nonetheless, we observed surprisingly frequent spontaneous apicoplast loss when culturing parasites in a medium bypassing the organelle's functions in the absence of any apicoplast inhibitors. We hypothesize that apicoplast loss is a common occurrence driven by frequent errors in the complex inheritance of the organelle[95–98]. While detrimental under physiological conditions, these errors are partially tolerated under bypass conditions, revealing their prevalence. An alternative interpretation of the observed spontaneous apicoplast loss is that the apicoplast rescue medium or expression of the MVA cassette itself might induce stress, triggering the loss of the organelle. However, this appears unlikely given that apicoplast-bearing parasites that express the MVA cassette grow equally well in the apicoplast rescue medium compared to control strains in the regular medium. Altogether, these findings hint that parasites in nutrient-rich environments quickly adapt to salvage metabolites typically produced by the apicoplast, potentially driving organelle loss.

The bypass of the apicoplast's functions in *T. gondii* opens new avenues for dissecting the organelle and ultimately targeting it. Given the extensive roles of the apicoplast in *T. gondii* and the utility and tractability of this model apicomplexan, this approach offers a considerable advancement over the bypass described for intraerythrocytic *P. falciparum*. Specifically, drug screens can be designed to compare the sensitivity of parasites dependent on the apicoplast or not, comparable to studies with intraerythrocytic *P. falciparum*[21–27]. In contrast to intraerythrocytic *P. falciparum*, such screens in *T. gondii* could also identify inhibitors of heme and FA synthesis, pathways that are dispensable in intraerythrocytic *P. falciparum*[29,30].

In addition, given the genetic tractability of *T. gondii*, comparative CRISPR-based genome-wide fitness screens can be conducted to identify all genes required for apicoplast maintenance and function[47,48,99]. Moreover, this tool can be adapted to interrogate the apicoplast's functions in other parasite stages such as liver stage *P. falciparum* or the encysted chronic stage of *T. gondii*, bradyzoites. A recent study revealed the apicoplast to be essential for bradyzoites[100], however, it remains unclear which pathways are needed during this stage. Understanding these pathways and developing strategies to target them is crucial for effective parasite control[101].

## Methods

### Parasite strains and cell culture
RH Δ*HXGPRT*Δ*KU80* (RH)[102] and RH DiCre Δ*HXGPRT*Δ*KU80* (DiCre)[49] strains were maintained in human foreskin fibroblasts (HFF, ATCC SCRC−1041), using Dulbecco's modified Eagle medium (DMEM, Dutscher L0104) supplemented with 5% fetal bovine serum (FBS, Capricorn Scientific, FBS−16A), 40 µg/ml gentamycin (Gibco, 15750-045) and additional glutamine (2 mM, Gibco, 25030) cultured in humidified incubators at 37 °C and 5% $CO_2$.

RH-MVA-HA$^{-Apico}$ parasites were maintained in apicoplast rescue medium (ARM), prepared as follows: DMEM as above was supplemented with 50 µM C16:0 (Sigma-Aldrich, P0500) and 50 µM C14:0 fatty acids (FAs, Sigma-Aldrich, M3128) coupled to FA-free bovine serum albumin (BSA) (Merck, 126575), 10 mM mevalonolactone (MVL, Sigma-Aldrich, M4667), 20 µM lysophosphatidic acid (0:0/14:0 LPA, Avanti, 857120 P) and 300 µM 5-aminolevulinic acid (5ALA, Sigma-Aldrich, A3785), and additional 2% FBS. Individual supplementations were performed with the above reagents as indicated for each experiment.

### Generation of transgenic strains
All inducible KD strains were generated through co-transfection of a CRISPR-Cas9 expression plasmid[47] with a guide RNA (Supplementary Data 1) targeting the 3'-UTR of each gene and a homology repair template encoding the 3-Ty tag, *LoxP* sites, U1 domain and the *HXGPRT* resistance cassette[49,50], amplified with the KOD primers by KOD PCR (Sigma-Aldrich, KMM−101NV) (Supplementary Data 1). Transfected parasites were selected in a medium containing mycophenolic acid (25 µg/ml) and xanthine (50 µg/ml) over one week and cloned by serial dilution in a 96-well plate. Downregulation of the gene of interest was achieved through treatment with rapamycin (50 nM).

Correct integration of the homology template at the desired location was assessed by PCR (GoTaq DNA Polymerase, Promega) on extracted genomic DNA (Promega Wizard DNA Extraction) using the integration primers listed in Supplementary Data 1, amplifying under the following conditions: 95 °C, 2 min; (95 °C, 15 s; 57 °C 15 s; 72 °C 1.5 min) × 30; 72 °C, 5 min on a SimpliAmp Thermal Cycler (Applied Biosystems).

iKD IspH-MVA-HA and RH-MVA parasites were generated in the iKD IspH and RH background, respectively, through transfection with a gRNA targeting the UPRT locus[57] (Supplementary Data 1) and an MVA cassette homology template following PCR amplification from a plasmid[45] with the primers listed in Supplementary Data 1. Parasites were selected for 7–10 days with 5 µM 5-fluorodeoxyuridine (FUDR) for UPRT-negative selection. Expression of the HA-tagged MVA protein in a clonal population was confirmed by IFA and Western blot.

Deletion of *UROS* and its replacement with a dihydrofolate reductase (DHFR) resistance cassette[103] was achieved using a 2guide RNA CRISPR/Cas9 strategy. Serial dilution cloning was performed under pyrimethamine treatment and clones lacking *UROS* were selected and validated using the guides and primers listed in Supplementary Data 1.

### Reagents and antibodies
Rapamycin (Sigma-Aldrich, 553210) was made up in DMSO at 1 mM and used at a final concentration of 50 nM (1:20,000). Actinonin (Sigma-Aldrich, A6671) stocks were made up at 40 mM in DMSO. Atovaquone (Sigma-Aldrich, PHR1591) and triclosan (Sigma-Aldrich, PHR1338) stocks were made up in DMSO at 10 mM. A clindamycin (Sigma-Aldrich, PHR1159) stock was made up at 10 mM in $H_2O$. All the above drugs were diluted in culture to achieve the final concentration as indicated for each experiment. The primary antibodies for IFAs were as follows: anti-GAP45 (1:10,000)[104], anti-actin (1:20)[105], anti-Ty (1:10, BB2), anti-Cpn60 (1:3,000)[51], anti-ATrx1 (1:3,000)[52], anti-HA (1:10, BB2). The secondary antibodies for IFAs were as follows: anti-mouse 488 (1:3000, Invitrogen A11001), anti-rabbit Alexa fluor 594 (1:3,000, Invitrogen A11012). The primary antibodies for western blot were as follows: anti-catalase (1:2000)[106], anti-Ty (1:10, BB2), anti-Cpn60 (1:3000), anti-HA (1:1000, Sigma-Aldrich,H6908), anti-IMC1 (1:1000)[107]. The secondary antibodies for western blot were as follows: anti-mouse HRP (1:3000, Sigma-Aldrich, A5278), anti-rabbit HRP (1:3000, Sigma-Aldrich, A8275).

### Western blots and solubility assays
Parasites were collected, resuspended in SDS-PAGE buffer (50 mM Tris-HCl, pH 6.8, 10% glycerol, 2 mM EDTA, 2% SDS, 0.05% bromophenol blue, and 100 mM dithiothreitol (DTT)) and boiled for 10 min. Proteins were separated by SDS-PAGE and transferred to a hybond ECL nitrocellulose membrane using a wet transfer system (Bio-Rad Laboratories). The membrane was blocked in 5% non-fat milk (0.05% Tween−20/PBS) and probed with primary antibodies overnight at 4 °C, washed, and then incubated with secondary antibodies. For solubility assays, freshly lysed parasites were collected, resuspended, and freeze-thawed 5 times in either PBS, PBS 1% Triton X−100, PBS 1 mM $Na_2CO_3$,

or 1% SDS. After incubating all samples on ice for 30 min, soluble and insoluble/pellet fractions (s and p, respectively) were separated by centrifugation (15,000 × g, 4 °C, 20 min), and processed as described for western blot samples above. All membranes were imaged by ChemiDoc Touch Imaging System (Bio-Rad Laboratories).

### Indirect immunofluorescence assays (IFAs)

Parasites grown in HFF cells on coverslips were fixed with 4% paraformaldehyde (PFA) and 0.05% glutaraldehyde for 10 min. Fixation was quenched with PBS containing 0.1 M glycine. The cells were permeabilized with PBS 0.2% Triton X-100 for 20 min, and then blocked with 2% BSA (0.2% Triton X-100/PBS) for 1 h, followed by incubation with the indicated primary antibodies. After washing with PBS 0.2% Triton X-100 for 3 times, the coverslips were incubated with the indicated secondary antibodies. Images were taken using a confocal microscope (Zeiss LSM700).

### Intracellular growth and apicoplast-loss quantification assays

HFF cells were grown on coverslips to confluency and infected with freshly egressing parasites for 24 h. For treatments of > 24 h (e.g., 72 h + Rapa), cells were pre-treated intracellularly for 48 h, before infecting host cells on coverslips and continuing to grow parasites under the indicated conditions for 24 h. For the drug treatment assays, cells were pre-treated for 12 h while growing intracellularly. To ensure that the results are not affected by an egress defect, these pre-treated parasites were syringe-lysed carefully and passed onto fresh host cells on coverslips, continuing the corresponding treatment over 24 h. IFAs were fixed and stained as described above, and cell numbers and apicoplast loss were quantified by staining with anti-GAP45 and anti-ATrx1 antibodies[52], respectively, to quantify the number of parasites per vacuole or the number of apicoplasts from 100 vacuoles. For two mutants (iKD FabG-Ty and iKD 248770-Ty) the apicoplast loss was additionally quantified using Cpn60[51] and DAPI staining, confirming the results obtained with ATrx1.

### Plaque assays

Freshly egressed parasites were inoculated on confluent HFF cells. When applicable, inoculation was performed ± rapamycin (Rapa) and metabolite supplementations as indicated. Parasites were grown for seven days before fixing with 4% PFA for 10 min and staining the monolayer with a crystal violet solution (12.5 g crystal violet, 125 ml ethanol mixed with 500 ml water containing 1% (w/v) ammonium oxalate) prior to washing with water. The plaques size was measured using the FindEdges and Segmentation (LevelSets) plugins in ImageJ (FIJI) software (10 plaques for each independent experiment).

### Transmission electron microscopy (TEM)

Samples for transmission electron microscopy were processes as described previously[108]. Briefly, infected HFF cells grown on round glass coverslips were fixed with a mixture of 2.5% glutaraldehyde and 2% PFA (Electron Microscopy Sciences) in 0.1 M sodium cacodylate buffer at pH 7.4 for 1 h at room temperature. After extensive washing with 0.1 M sodium cacodylate buffer, the pH 7.4 sample was post-fixed with reduced 1% osmium tetroxide (Electron Microscopy Sciences) with 1.5% potassium ferrocyanide in 0.1 M sodium cacodylate buffer, pH 7.4 for 1 h and immediately followed by 1% osmium tetroxide alone in the same buffer for 1 h. After two washes in double distilled water for 5 min each wash samples were *en block* stained with aqueous 1% uranyl acetate (Electron Microscopy Sciences) for 1 h or overnight at 4 °C. After 5 min wash in double distilled water cells were dehydrated in graded ethanol series (2 × 50%, 70%, 90%, 95%, and 2 × absolute ethanol) for 10 min each wash and infiltrated with graded series of Durcupan resin (Electron Microscopy Sciences)

diluted with ethanol at 1:2, 1:1, 2:1 for 30 min each. Cells were infiltrated twice with pure Durcupan for 30 min each and with fresh Durcupan resin for an additional 2 h. Finally, coverslips with cells facing down were placed on 1 mm thick teflon rings filled with resin and placed on a glass slide coated with mold separating agent (Glorex) and polymerized in the oven at 65 °C for 24–48 h. The glass coverslip was removed from the cured resin disk by alternate immersion into hot (60 °C) water and liquid nitrogen until the glass parted. A laser microdissection microscope (Leica Microsystems) was used to select suitable areas and to outline their positions on the resin surface to cut out from the disk using a single-edged razor blade and glued with superglue (Ted Pella) to a blank resin block. The cutting face was trimmed using a Leica Ultracut UCT microtome (Leica Microsystems) and a glass knife. 70 nm ultrathin serial sections were cut with a diamond knife (DiATOME) and collected onto 2 mm single slot copper grids (Synaptec, Ted Pella) coated with Formvar plastic support film.

Sections were examined by Tecnai 20 TEM (FEI) operating at an acceleration voltage of 80 kV and equipped with a side-mounted MegaView III CCD camera (Olympus Soft-Imaging Systems) controlled by iTEM acquisition software (Olympus Soft-Imaging Systems) at the Electron Microscopy Facility (PFMU) at the Department of Medicine at the University of Geneva.

### GC-MS analyses

**Cell harvest and labeling of extracellular parasites.** Parasites were treated for the indicated duration in regular DMEM (normal medium, NM) or apicoplast rescue medium (ARM) with or without rapamycin (50 nM) and/or heavy water (deuterium oxide, $^2H_2O$, 3.6% v/v; Apollo Scientific, DE50K). Freshly egressing parasites were harvested by gently removing the culture medium, quenching the metabolism through the addition of ice-cold PBS, scraping the monolayer, and extracting parasites by passage through a 26 G needle. The cell suspension was filtered (3 μm exclusion size, Merck-Millipore, TSTP04700) to remove host cell debris and collected into 15 ml conical tubes. Parasites were pelleted (2000 × g, 4 °C, 25 min) and washed two more times with ice-cold PBS. Residual PBS was removed, and pellets of $10^8$ parasites stored at − 80 °C until metabolite extraction or harvested parasites were resuspended and incubated for 5 h in glucose-free DMEM (Gibco, 11966025), supplemented with 5% dialyzed FBS (Pan Biotech P30−2102) and medium containing 10 mM U-$^{13}C_6$-glucose (Sigma-Aldrich, 389374) or natural abundance glucose (control). For these extracellularly labeled parasites, the medium was pre-incubated in the cell's incubator, and samples were intermittently agitated during the incubation period to facilitate gas exchange and ensure even cell suspension. Following the intended labeling duration, cells were pelleted and washed as described above. FAs of these parasites were extracted, derivatized, and analyzed as outlined below.

**Metabolite extraction.** Metabolites were extracted as previously described but omitting any heating step[109]. Briefly, parasite pellets were thawed on ice prior to the addition of 50 μl chloroform (containing 50 μM pentadecanoic acid as internal standard, Sigma-Aldrich, P6125) followed by 200 μl methanol:ultrapure water (3:1, containing scyllo inositol as an internal standard, 5 μM, Sigma-Aldrich, I8132). Samples were vigorously vortexed before centrifugation (20,000 × g, 4 °C, 10 min). The supernatant was transferred to a new vial containing 100 μl ice-cold ultrapure water. Samples were vortexed and spun (20,000 × g, 4 °C, 10 min). The lower organic phase (apolar, 50 μl) and the upper, polar phase (300 μl) were processed further as outlined below.

**Derivatization and GC-MS analysis.** The organic phase was dried, and FAs reconstituted in 60 μl chloroform: methanol (2:1) containing 3% (v/v) 3-(trifluoromethyl) phenyltrimethylammonium hydroxide (Tokyo

Chemical Industry, TCI, T0961) to convert FAs to their methyl esters (FAMEs). FAMES were analyzed by GC-MS on an 8890 GC System (Agilent) equipped with a DB5 capillary column (J&W Scientific, 30 m, 250 μm inner diameter, 0.25 μm film thickness), with a 10 m inert duraguard. The GC was connected to a 5977B GC/MSD in electron impact (EI) mode equipped with a 7693 A autosampler (Agilent). The GC-MS settings were as follows: Inlet temperature: 270 °C, MS transfer line temperature: 280 °C, MS source temperature: 230 °C and MS quadrupole temperature: 150 °C, oven temperature gradient:80 °C (2 min); 80 °C to 140 °C at 30 °C/min; 140 °C to 250 °C at 5 °C/min; 250 °C to 310 °C at 15 °C/min; 310 °C for 2 min. Samples were analyzed in scan mode ($m/z$ 70–700) as well as in selected ion monitoring (SIM) mode, for the $^2H_2O$-labeled samples, selecting the molecular ion M0 and the M + 1 as follows: myristic acid (C14:0, $m/z$ 242, 243), palmitic acid (C16:0, $m/z$ 270, 271), oleic acid (C18:1, $m/z$ 296, 297), stearic acid (C18:0, $m/z$ 294, 295). These fatty acids were identified based on the retention time and ion spectra of authentic standards. The excess molar enrichment of the M + 1 was calculated based on the SIM acquisition[54]. The isotopologue abundance measurement following U-$^{13}C_6$-glucose labeling was based on scan mode acquisition and corrected for natural abundance as described previously[110].

The polar phase was dried sequentially in a centrifugal evaporator and further dried and concentrated through the addition of methanol. The dried metabolite extract was derivatized through the addition of 25 μl pyridine containing methoxyamine hydrochloride (Sigma-Aldrich, 226904) at 20 mg/ml followed by incubation at room temperature overnight. The next day, 25 μl N, O-Bis(trimethylsilyl, TMS) trifluoracetamid 99% (Supelco, B-023) were added, and samples vortexed and analyzed, using the same instrument and inlet and MS temperature settings as above. The following oven gradient was used for separation of polar metabolites: 70 °C (1 min); 70 °C to 295 °C at 12.5 °C/min; 295 °C to 320 °C at 25 °C/min; 320 °C for 2 min and operating in scan mode ($m/z$ 70–700) with a 5.5 min solvent delay. Polar metabolites were identified based on the analysis of authentic standards or reliable predictions (NIST library, NIST MS Search 2.4, > 60% confidence and manual curation). Note that the analysis of polar metabolites also retrieved TMS derivatives of fatty acids and other lipidic compounds, either due to an incomplete separation of the phases or due to their association with more polar compounds prior to dissociation during derivatization. All GC-MS data was analyzed using MassHunter (Quantitative Analysis and Qualitative Analysis 10.0, Agilent) and Excel (Microsoft). For the metabolite profiling, compounds were quantified based on the abundance of a suitable quantifier ion at its specific retention time with a left and right RT delta of 0.07 min. The quantifier ion was selected based on its abundance and specificity, avoiding any overlap with closely eluting metabolites. The list of quantifier ions and retention times are provided in the Supplementary Data 3. An internal standard was included in the analysis for normalization (scyllo-inositol, eluting at 14.75 min). However, due to differences in the overall signal intensities of parasite culture in NM or ARM, the normalization to the total ion chromatogram signal intensity was found to be more reliable here. For each metabolite, the abundances were determined relative to the given reference, which was set to a relative abundance of 1. All results related to this analysis can be found in Supplementary Data 2–4. The raw data corresponding to all the above metabolomic analyses has been deposited[111,112].

### Proteomic analyses

**Parasite treatment and harvest.** RH, RH-MVA, and RH RH-MVA-HA$^{-Apico}$ parasites were grown over 48 h in a normal medium or in ARM. Freshly egressing parasites were harvested as described above but with the final wash carried out in PBS, containing protease inhibitor (Pierce, Protease Inhibitor EDTA-free, A32965). Cell pellets were snap-frozen and stored at −80 °C.

**Sample preparation.** Sample preparation and proteomics analyses was performed by the Proteomics Core Facility at the University of Geneva. Cell pellets were resuspended in 50 μl of 0.1% RapiGest Surfactant (Waters) in 50 mM ammonium bicarbonate (AB). Samples were heated for 5 min at 100 °C. Lysis was performed by sonication (6 × 30 sec.) at 70% amplitude and 0.5 pulse. Samples were kept for 30 sec on ice between each cycle of sonication. Samples were centrifuged for 10 min at 14,000 × g. Protein concentration was measured by Bradford assay, and 50 μg of each sample was subjected to protein digestion as follows: sample volume was adjusted to 100 μl with 0.1% RapiGest in 50 mM AB. 2 μl of Dithioerythritol (DTE) 50 mM were added and the reduction was carried out at 37 °C for 1 h. Alkylation was performed by adding 2 μl of iodoacetamide 400 mM during 1 h at room temperature in the dark. Overnight digestion was performed at 37 °C with 10 μl of freshly prepared trypsin (Promega; 0.1 μg/μl in 50 mM AB). To remove RapiGest, samples were acidified with trifluoroacetic acid (TFA), heated at 37 °C for 45 min, and centrifuged for 10 min at 17,000 × g. Supernatants were then desalted with a C18 microspin column (Harvard Apparatus) according to the manufacturer's instructions, completely dried using a centrifugal evaporator, and stored at −20 °C.

**ESI-LC-MSMS.** Samples were dissolved at 1 μg/μl with loading buffer (5% acetonitrile, 0.1% formic acid). Biognosys iRT peptides were added to each sample, and 2 μg of peptides were injected into the column. Liquid chromatography-electrospray ionization-tandem mass spectrometry (LC-ESI-MS/MS) was performed on an Orbitrap Fusion Lumos Tribrid mass spectrometer (Thermo Fisher Scientific) equipped with an Easy nLC1200 liquid chromatography system (Thermo Fisher Scientific). Peptides were trapped on an Acclaim pepmap100, C18, 3 μm, 75 μm × 20 mm nano trap-column (Thermo Fisher Scientific) and separated on a 75 μm × 500 mm, C18 ReproSil-Pur (Dr. Maisch GmBH), 1.9 μm, 100 Å, home-made column. The analytical separation was run for 135 min using a gradient of $H_2O$/formic acid 99.9%/0.1% (solvent A) and acetonitrile/$H_2O$/formic acid 80.0%/19.9%/0.1% (solvent B). The gradient was run from 8% B to 28% B in 110 min, then to 42% B in 25 min, then to 95% B in 5 min with a final stay of 20 min at 95% B. The Flow rate was of 250 nl/min, and the total run time was 160 min. Data-independent acquisition (DIA) was performed with MS1 full scan at a resolution of 60,000 (full width at half maximum, FWHM) followed by 30 DIA MS2 scans with fix windows. MS1 was performed in the Orbitrap with an automatic gain control (AGC) target of 1 × 106, a maximum injection time of 50 ms, and a scan range from $m/z$ 400 to 1240. DIA MS2 was performed in the Orbitrap using higher-energy collisional dissociation (HCD) at 30%. Isolation windows were set to $m/z$ 28 with an AGC target of 1 × 10⁶ and a maximum injection time of 54 ms.

**Data analysis.** DIA raw files were loaded into Spectronaut v.18 (Biognosys) and analyzed by directDIA using default settings. Briefly, data were searched against the *Toxoplasma gondii* GT1 database (ToxoDB, release 57, 8460 entries). Trypsin was selected as the enzyme, with one potential missed cleavage. Variable amino acid modifications were oxidized methionine. The fixed amino acid modification was carbamidomethyl cysteine. Both peptide precursor and protein false discovery rates (FDRs) were controlled at 1% (Q value < 0.01). Single Hit Proteins were excluded. For quantitation, the protein label free quantification (LFQ) method was set to "QUANT 2.0", "only protein group specific" was selected as proteotypicity filter and normalization was set to "automatic". The quantitative analysis was performed with Spectronaut, proteins were considered to have significantly changed in abundance with a Q-value < 0.05 and an absolute fold change FC≥ |1.5| (log2FC ≥ |0.58| ). Data was further analyzed using Excel (Microsoft) and GraphPad Prism. All results

related to this analysis can be found in Supplementary Data 5. The corresponding raw data has been deposited[111].

### Transcriptomic analyses

**Parasite harvest.** RH-MVA and RH RH-MVA-HA[−Apico] parasites ($n = 4$) were grown over 48 h in the apicoplast rescue medium (ARM). Freshly egressing parasites were harvested as described above for the metabolomics samples and cell pellets snap-frozen and stored at − 80 °C.

**Sample preparation and quality control.** Total RNA was obtained using a Trizol (Life Technologies) extraction, as previously described[113]. The obtained RNA was resuspended in 50 µl RNase-free water, snap-frozen, and stored at −80 °C. The average RNA concentration across the 8 samples was found to be 698 ng/µl, as determined by Qubit RNA IQ Assay (Thermo Fisher Scientific) while the average RNA integrity number, determined by Bioanalyzer (Agilent), was found to be 9.75.

**Illumina sequencing.** Sequencing was performed using the NovaSeq 6000 Sequencing System (Illumina) at the iGE3 Genomics platform, University of Geneva. Sequencing quality control was performed with the FastQC quality control tool, finding a mean quality score of 36 (Babraham Bioinformatics). The fastq files were mapped to the *Toxoplasma gondii* GT1 Ensembl reference with BWA mem v.0.7.17[114]. The average number of mapped reads was over 80,000,000, per sample, corresponding to a read mapping of 99.19%. The table of counts with the number of reads mapping to each gene feature of the *Toxoplasma gondii* GT1 Ensembl reference gff3 file was prepared with HTSeq v0.9.1[115]. After normalization, the poorly detected genes were filtered out, and out of 8460 total genes, 7242 genes with a count above 10 were kept for analysis. The differential expression analysis was performed with the statistical analysis R/Bioconductor package edgeR1.38.4[116] with a multiple testing Benjamini and Hochberg correction FDR of 5% and a fold change threshold of 2. All results related to this analysis can be found in the Supplementary Data 6. The corresponding raw data has been deposited[112].

### Animal experimentation

All animal experiments were conducted with the authorization number GE125A, according to the guidelines and regulations issued by the Swiss Federal Veterinary Office. Mice were housed at the University of Geneva, Centre Médical Universitaire animal facility in a room with a day/night cycle of 12 h/12 h and constant ambient temperature of 22 °C and 35% humidity. Virulence assays were performed by intraperitoneal injection of 1000 freshly lysed RH parasites in 7-week-old female B6CBAF1/J mice. Mice were monitored daily and sacrificed at the onset of signs of acute infection (ruffled fur, difficulty moving, isolation).

### Statistical analyses

Statistical differences between the two groups were tested by unpaired 2-tailed Student's *t* test with SPSS20 statistical software or One-way ANOVA followed by Tukey's pairwise comparison, as indicated for each experiment. All *p*-values are given, and statistical significance was defined as *p*-value < 0.05. All experiments were repeated at least three times. All images shown (IFAs, plaque assays, western blots, etc.) depict representative results from one of three or more experiments.

### Reporting summary

Further information on research design is available in the Nature Portfolio Reporting Summary linked to this article.

## Data availability

The authors declare that the data supporting the findings of this study are available within the paper and its supplementary information files. The raw data associated with the proteomic and metabolomic analyses (metabolite profiling and $^2H_2O$ labeling analysis of fatty acids) have been deposited in the Yareta (Geneva) repository, which complies with the FAIR principles, under the URL: https://doi.org/10.26037/yareta:kc446nwxifbvtb5z7l2f6tayga[111]. Additional raw data pertaining to the $^{13}$C-labeling analysis of fatty acids and the transcriptomic data have been deposited in the Yareta repository under the URL: https://doi.org/10.26037/yareta: lb356tpupbcfteq4t4ikalhapq[112]. Source data are provided in this paper.

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

## Acknowledgements

We thank Prof. Bang Shen for kindly sharing the MVA plasmid. In addition, we thank Dr. Fedor Bezrukov for his expert bioinformatical support, as well as the Proteomics Core Facility and the iGE3 Genomics platform at the Centre Médical Universitaire (University of Geneva) for their help with the proteomics and transcriptomics analyses, respectively. This work is supported by the grants #TMAG-3_216166 and IZLIZ3_200277 awarded by the Swiss National Science Foundation (https://www.snf.ch/en) to D.S.-F., as well as the Key Project by the National Natural Science Foundation of China, 82330072, awarded to H.-J.P. J.K. is supported through funding by a generous donor advised by CARIGEST SA (https://carigest.ch/en), acquired by D.S.-F. This work was supported by a grant from the Novartis Foundation for Medical Biological Research (#22C164, http://www.stiftungmedbiol.novartis.com/old-home.html) and funding from the Foundation Gertrude von Meissner (http://www.medecine.unige.ch/lafaculte/commissions/commissions_temporaires/gvm/) awarded to J.K. The funders had no role in study design, data collection and analysis, decision to publish, or preparation of the manuscript. We are grateful for the invaluable resources provided by VEuPathDB.

## Author contributions

M.C., D.S.-F., and J.K. conceived the study; M.C. and J.K. performed and interpreted the experimental work, with the support of S.G.K., A.B., O.V.R., and B.M.; D.S.-F., H.-J.P., and J.K. supervised the research. M.C. and J.K. wrote the paper with the support of D.S.-F., S.G.K., A.B., O.V.R., B.M., and H.-J.P.

## Competing interests

The authors declare no competing interests.
