## [Transparent Peer Review file · Nature Communications]

Dissecting apicoplast functions through continuous cultivation of *Toxoplasma gondii* devoid of the organelle

Corresponding Author: Dr Joachim Kloehn

Version 0:

Reviewer comments:

Reviewer #1

(Remarks to the Author)

In this manuscript, Chen and colleagues take a top down approach to characterize four metabolite-generating pathways in the apicoplast of the parasite *Toxoplasma gondii*. In doing so, they are able to bypass each of those pathways, and ultimately generate apicoplast-less (*T.gondii*-Apico in the text) parasites.

The apicoplast is an organelle only found in apicomplexans, a group of protozoan parasites which also includes *Plasmodium falciparum*, the causative agent of malaria. In the blood stages of this parasite, the absence of the apicoplast can be bypassed with the addition of isopentenyl pyrophosphate (IPP) if added to the culture medium. In *T. gondii* the apicoplast plays other roles, and Chen and colleagues carefully dissected the role of four output-generating pathways considered to be essential for apicoplast function: i) de novo fatty acid biosynthesis, ii) synthesis of lysophosphatidic acid, iii) heme synthesis and iv) IPP synthesis. Using a conditional knockdown approach, they confirm the localization of one or more proteins in each of those pathways to the apicoplast. They also demonstrate that they are essential for parasite survival and assess if their depletion triggers a loss of the apicoplast. Importantly, they demonstrate that the growth defect observed upon protein depletion can be bypassed by the addition of precursors or intermediates for each metabolic pathway, respectively: i) fatty acids, ii) lysophosphatidic acid, iii) 5-aminolevulinic acid (5ALA) and iv) mevalonolactone (MVL).

Next the use the drug actinonin, known to disrupt apicoplast biogenesis in *P. falciparum*, to trigger apicoplast loss and death in *T. gondii*. Actinonin seems to target a membrane protease called FtsH1 and the authors also generate a conditional knockdown strain to study this protease. Depletion of FtsH1 also triggers apicoplast loss. The authors then aimed to rescue the apicoplast loss caused via actinonin. They first generated a strain of *T. gondii* expressing a mevalonate bypass cassette (MVA), consisting of a multifunctional polypeptide of human and bacterial origin able to covert MVL to IPP. This strain is then treated with actinonin and cultured in either regular media or apicoplast-rescue media (ARM), which contains the metabolites able to bypass the four output-generating pathways in the apicoplast mentioned in the previous paragraph. This *T.gondii*-Apico can be maintained in culture even though they seem to grow slowly compared to parasites with an apicoplast. Importantly, Chen and colleagues show that the apicoplast-less parasites are less sensitive to drugs that target the apicoplast.

Finally, the authors demonstrate that numerous nuclear encoded proteins, many of them predicted to localize to the apicoplast, are downregulated in the apicoplast-less *T. gondii*. They further validate the localization of five of those downregulated proteins, including one that did not have any assigned localization. This last protein is further characterized as a Coccidia-specific apicoplast protein.

Comments:

The generation of an apicoplast-less strain of *T. gondii* is a major accomplishment and tour de force for the field. This achievement will be invaluable in studying the apicoplast, its essential proteins, and its drug susceptibility. However, the manuscript is extremely dense and hard to follow, so I would recommend the authors to reorganize it. I felt like the generation and characterization of the apicoplast-less parasite is buried deep in the paper given all the other data and results that the authors put in the manuscript. I had an extremely hard time following the switching between the main figures and the supplementary figures. I would therefore ask the authors to reorganize their data to have all the results of the main figures together, and then all the supplementary results (particularly for figures 1 to 3).

I also was not sure if I fully understood the point of the experiments targeting FtsH1. The authors mention that they performed

the first functional study FtsH1, but I think this is an overstatement and this should be reframed as a preliminary functional study. The authors do not mention that the protein was already localized to the apicoplast in 2009 by Kamataki et al. (PMID: 17822404). Then, they characterize the depletion of the protein, which leads to apicoplast loss. Given what we knew of the protein from Plasmodium and Toxoplasma, the results from the authors are not surprising at all. Yet, the results concerning FtsH1 are displayed as a main figure, but the authors never build upon those results again. To me, this portion of the manuscript seems disconnected from the main story (Which can be divided into the strategy to bypass the four output-generating pathways, the generation of the apicoplast-less parasites via actinonin treatment and their subsequent preliminary characterization), so I would recommend to move them to the supplementary material. To me, a functional study will go beyond what the authors did to fully understand the role of this metalloprotease, and will probably be a standalone paper.

Here is a list of other elements that, in my opinion, need to be addressed to improve to quality of the paper.

- Figure 1 is extremely dense and complex, it will be better only highlight the proteins studied by the authors and just mention each of the pathways. We do not need the names and abbreviation of all the enzymes and this stage.
- Line 290: "its effectiveness (of the MVA cassette) remained unclear". What do the authors mean? Can they clarify this point? In the referenced paper (PMID: 36449552), it is shown that adding MEV to the media rescued the deficiency of the triosephosphate isomerase 2 mutants generated (Figure 6 of the aforementioned paper). That in my mind seems that its effectiveness was demonstrated?
- Line 411-412: parasites treated with actinonin formed plaques close to normal size when cultured in ARM. This is an overstatement, the plaques are less than half as big. They are plaques there, but not at all close to the size of the parental strain. Please reword accordingly.
- Why did the authors choose to provide complete ARM with FA and LPA even though it seemed that one could be sufficient? They do not provide any reasoning behind that decision (line 421).
- Regarding the generation of the apicoplast-less parasites, is this phenotype reversible? It is not clear in the paper if the authors need to keep the parasites in the presence of actinonin and ARM to maintain the phenotype or if the actinonin treatment was only performed for 20 days (line 423).

Minor comments:

- In all the figures with a Western Blot, there is no indication of how many times the WB were repeated in either the legend or in the M&M section.
- In all the figures with an intracellular growth assay, the colors should be the same to avoid any confusion. For instance, in figure 2m choose the same color scheme (blue or orange) instead of blue and orange.
- Why do we see several bands in figure S5c? The expression of the MVA cassette did not show that pattern when expressed in the iKD IspH-MVA-HA strain (figure 3h) and the authors do not offer any explanation for this. In the same vein, why don't we see both premature and mature forms of Cpn60 in figure 5g like we see in figure 4i for RH parasites?
- Figure 5k: why are the plaque assays in another color? They look very different from the plaque assays from the rest of the figure.
- Figure S5i: several panels seems to be at a very low res.

Reviewer #2

(Remarks to the Author)

In the manuscript titled "Dissecting apicoplast functions through continuous cultivation of *Toxoplasma gondii* devoid of the organelle" by Chen et al. have attempted to expand the knowledge of apicoplast functions beyond the already extensive literature by knocking out the apicoplast completely in the model apicomplexan *Toxoplasma gondii*. The authors achieved apicoplast removal by first sequentially disrupting the four essential metabolic pathways housed by the organelle and then attempting to bypass each pathways either by directly supplying the end products of the pathways in the culture medium or by expressing enzymes that can convert supplemented substrates into the products/outputs of the essential pathways. In each case this work relies on robust knowledge that was already available in the field.

After these validations, the authors chooses to knock out the apicoplast by actinonin treatment, which inhibits the FtsH1 metalloprotease required for apicoplast biogenesis. This apicoplast-knocked-out cell line is maintained in continuous culture by supplementing the medium through the previously known rescue or bypass methods validated at the beginning of the work.

The deletion of the apicoplast has previously been accomplished in the intraerythrocytic stage of the closely related Plasmodium species and thus the conceptual progress minor to the field. However, this study presents the first instance of such an undertaking in *Toxoplasma*, which has high potential to be valuable for *Toxoplasma* researchers and to aid in a detailed understanding of the biogenesis and function of this important organelle. Yet, the current work does not present novel findings. The comprehensive analysis of all the cell lines, as well as the quality of data generated by the authors, is commendable. Unfortunately, the findings do not yet add much upon existing knowledge to further elucidate the biology of the apicoplast in *Toxoplasma*.

Major points:

1. While the loss of the apicoplast in the clonally selected *T. gondii*-Apico cell line has been thoroughly validated using

immunofluorescence assays (IFA) to stain various apicoplast markers, PCR for genomic fragments, and electron microscopy (EM), the same cannot be asserted for other cell lines used as validation of previous work, where partial or extreme apicoplast loss is claimed. The apicoplast, with its multiple membrane-bound compartments and resident genome, may not be entirely absent if only one protein (e.g., ATRx1) is missing from a specific compartment. As long as the organelle's genome is present, the organelle itself cannot be considered lost. Therefore, it is recommended that the authors validate apicoplast loss in the various knockdown cell lines using PCR for apicoplast genomic regions and, if possible, with IFA for an apicoplast marker in a different compartment of the organelle.

2. For iKD FabG, ATS1, and LipA, the authors claim that they are bypassing the essential role of these enzymes by supplementing the medium with the respective end products. However, according to plaque assay results, growth defects are only partially rescued; it is also unclear whether these cell lines with supplemented medium can be grown indefinitely in culture. With these points in mind, it would be wrong to claim that these pathways have been bypassed—rather, in this case, the defect has simply been partially rescued. It is notable that this rescue effect for both FA and lipid precursor synthesis and accessory pathways is not a new result and has already been shown in several publications before, as mentioned in the manuscript.

3. In the third results section, the authors aim to further characterize the function of FtsH1 in *T. gondii*. FtsH1 has been previously shown to localize in the apicoplast, is important for apicoplast biogenesis in *Plasmodium* and *Toxoplasma*, and is predicted to be involved in membrane protein processing in the apicoplast. Although the authors performed mass spectrometry analysis on the FtsH1-depleted cell line, they did not elucidate the protein's function beyond what was already known. Furthermore, there is insufficient explanation or analysis regarding the upregulation or downregulation of certain proteins, particularly why most of these are ER-resident proteins.

4. Can the authors confirm that some of these upregulated proteins upon FtsH1-depletion are substrates of FtsH1 through biochemical analyses? Can it be verified that the proteins accumulating in the iKD-FtsH1-Ty cell lines have premature peptides or have processing defects?

5. Additionally, the statement about the interaction between Cpn60 and FtsH1 is quite handwavy without further experimental evidence. While the authors compare this to plants, it should be noted that, unlike *Toxoplasma*, which has only one FtsH protein, plants can contain several FtsH proteins—12 in case of *Arabidopsis*. Only one of these, FtsH11, localizes in the inner envelope membrane, has been shown to interact with Cpn60 in its proteolytically inactive mutant form, questioning the biological relevance of this "artefactual" interaction. Comparing this to justify the downregulation of Cpn60 in *Toxoplasma* seems tenuous without further exploration and evidence. The authors could perform immunoprecipitation of the FtsH1-Ty tagged cell line to identify the interaction partners of FtsH1. Likewise, are the five downregulated proteins all interacting partners of FtsH1?

6. A potentially exciting outcome of maintaining an apicoplast-less *T. gondii* cell line would be to test whether inhibitors/compounds have specific targets in the apicoplast, as also pointed out by the authors. In this case, the authors tested doxycycline to verify their claims, as it should target the ribosome in the apicoplast. However, there is literature that suggests doxycycline also targets mitochondrial ribosomes. It would be helpful if the authors chose an inhibitor which is not actinonin that was used to induce apicoplast loss, but for which the target is known to be exclusive to the apicoplast (e.g., clindamycin), meaning the parasites would be completely insensitive to its use in culture.

7. As the apicoplast is known to be a metabolic hub for the parasite, it is disappointing to see that the authors haven't provided an in-depth analysis of the major changes in metabolites using the metabolomics data. As presented the data is inconclusive or hard to interpret. This seems like a missed opportunity to delve into insights that the field does not already hold about apoplast function.

Minor points:

1. Line 897, "plague" should be "plaque"
2. Line 417 and Suppl. Fig 5d – "MVL" instead of "MEV"

Reviewer #3

(Remarks to the Author)

Version 1:

Reviewer comments:

Reviewer #1

(Remarks to the Author)

The authors have greatly improved the quality of the manuscript. They have addressed all my comments, and by restructuring the paper they have enhanced its readability and focus. Congratulations!

I have only a few minor comments that can be solved before publication:

1) Typo in line 347: "in" is repeated two times

2) The generation of the RH-MVA-HA–Apico Δ UROS lacks one detail. This gene is essential in normal culture conditions, so where all the experiments to generate the strain performed in ARM media? I am assuming so since the strain is not viable in NM for extended periods of time, but it should be mentioned in the M&M section. Of note, it is interesting to see how an essential gene was now being treated as an indispensable gene (by simply replacing it with an antibiotic resistant cassette) in the RH-MVA-HA–Apico background.

3) Line 627: TGGT1_248770 has a hydrophobic region near its C terminus. The authors should mention how they found this region. Was it by using data available on ToxoDB? Or via other means?

4) I believe the solubility assay protocol is missing from the Materials and Method Section. I was able to find a protocol for all the other procedures in the manuscript except for that one.

Reviewer #2

(Remarks to the Author)

In this revised version of the manuscript, the authors have responded satisfactorily to all our previous queries. Therefore, I recommend this article to be accepted for publication.

Reviewer #3

(Remarks to the Author)

Reviewer #4

(Remarks to the Author)

I was asked by the editor to focus on the transcriptomic data presented in this manuscript (Specifically, the data presented in Fig. 7d, and supplementary data included in Table 6) and hence my comments are largely focused on this section of the manuscript.

The authors had previously looked at the protein levels in parasites with (RH-MVA-HA) and without the apicoplast (RH-MVA-HA–Apico). The data from these experiments had indicated that nuclear encoded apicoplast protein levels were downregulated in parasites lacking apicoplast. To determine if the downregulation of proteins is due to regulation at the transcriptional level or degradation in the absence of proper trafficking, authors performed RNA seq analysis.

The data presented in Fig.7d is compiled from three experiments is robust and the authors correctly interpret the data. Since most of the proteins that were found to be downregulated at the protein level, did not show significant changes in their RNA transcripts number suggests that the decreased abundance is mainly due to impaired stability. Hence the authors interpretation is indeed spot on.

Additionally, after careful reading of the revised manuscript, I feel that authors have adequately addressed all the reviewers comments.

Point-by-Point Response to Reviewers

We thank the reviewers for their valuable feedback and appreciate the opportunity to revise our manuscript accordingly. We have addressed all points as detailed below. Our answers are written in red.

Reviewer #1

In this manuscript, Chen and colleagues take a top down approach to characterize four metabolite-generating pathways in the apicoplast of the parasite *Toxoplasma gondii*. In doing so, they are able to bypass each of those pathways, and ultimately generate apicoplast-less (*T.gondii*-Apico in the text) parasites.

The apicoplast is an organelle only found in apicomplexans, a group of protozoan parasites which also includes *Plasmodium falciparum*, the causative agent of malaria. In the blood stages of this parasite, the absence of the apicoplast can be bypassed with the addition of isopentenyl pyrophosphate (IPP) if added to the culture medium. In *T. gondii* the apicoplast plays other roles, and Chen and colleagues carefully dissected the role of four output-generating pathways considered to be essential for apicoplast function: i) de novo fatty acid biosynthesis, ii) synthesis of lysophosphatidic acid, iii) heme synthesis and iv) IPP synthesis. Using a conditional knockdown approach, they confirm the localization of one or more proteins in each of those pathways to the apicoplast. They also demonstrate that they are essential for parasite survival and assess if their depletion triggers a loss of the apicoplast. Importantly, they demonstrate that the growth defect observed upon protein depletion can be bypassed by the addition of precursors or intermediates for each metabolic pathway, respectively: i) fatty acids, ii) lysophosphatidic acid, iii) 5-aminolevulinic acid (5ALA) and iv) mevalonolactone (MVL).

Next they use the drug actinonin, known to disrupt apicoplast biogenesis in *P. falciparum*, to trigger apicoplast loss and death in *T. gondii*. Actinonin seems to target a membrane protease called FtsH1 and the authors also generate a conditional knockdown strain to study this protease. Depletion of FtsH1 also triggers apicoplast loss. The authors then aimed to rescue the apicoplast loss caused via actinonin. They first generated a strain of *T. gondii* expressing a mevalonate bypass cassette (MVA), consisting of a multifunctional polypeptide of human and bacterial origin able to convert MVL to IPP. This strain is then treated with actinonin and cultured in either regular media or apicoplast-rescue media (ARM), which contains the metabolites able to bypass the four output-generating pathways in the apicoplast mentioned in the previous paragraph. This *T.gondii*-Apico can be maintained in culture even though they seem to grow slowly compared to parasites with an apicoplast. Importantly, Chen and colleagues show that the apicoplast-less parasites are less sensitive to drugs that target the apicoplast.

Finally, the authors demonstrate that numerous nuclear encoded proteins, many of them predicted to localize to the apicoplast, are downregulated in the apicoplast-less *T. gondii*.

They further validate the localization of five of those downregulated proteins, including one that did not have any assigned localization. This last protein is further characterized as a Coccidia-specific apicoplast protein.

Comments:

The generation of an apicoplast-less strain of *T. gondii* is a major accomplishment and tour de force for the field. This achievement will be invaluable in studying the apicoplast, its essential proteins, and its drug susceptibility. However, the manuscript is extremely dense and hard to follow, so I would recommend the authors to reorganize it. I felt like the generation and characterization of the apicoplast-less parasite is buried deep in the paper given all the other data and results that the authors put in the manuscript. I had an extremely hard time following the switching between the main figures and the supplementary figures. I would therefore ask the authors to reorganize their data to have all the results of the main figures together, and then all the supplementary results (particularly for figures 1 to 3).

We sincerely thank the reviewer for this suggestion. In response, we have significantly restructured the manuscript in several ways:

1. **Panel Restructuring:** We have reorganized **Figs. 2 and 3** to group all plaque assays, immunofluorescence assays (IFAs), western blots, etc., within the same panels. This change allows for a more cohesive presentation of the data, minimizing the need to switch between the main and supplementary figures.
2. **Data Redundancy Removal:** We have eliminated redundant data concerning the heme synthesis pathway. Previously, we described two mutants in the heme synthesis pathway, targeting PBGD and UROS; however, both behaved identically. To streamline our data presentation, we have retained only the PBGD data, which also aligns with our approach of presenting a single mutant for all other targeted pathways. This adjustment has reduced the supplementary data associated with **Fig. 3**.
3. **FtsH1 Data Reduction and Moving:** We have also removed certain data related to FtsH1 (details and reasons for this are provided below) and moved its partial characterization to the final figure (**Fig. 8**). This adjustment has allowed us to sharpen our focus on the metabolic pathways and their bypass mechanisms. Consequently, the generation and characterization of apicoplast-less parasites have been moved to the main **Fig. 4** (previously Fig. 5), shifting it to the center of this manuscript.

4. **Other structural Revisions:** Substantial additional changes have been made to the second half of the manuscript (fully revised **Figs. 4-8**) to address other reviewers' comments and accommodate new data, as outlined below.

Overall, we believe that these revisions have enhanced the clarity and focus of the paper, making the data more accessible to the readers. We hope the reviewer finds these changes satisfactory and we appreciate the guidance that led to this improvement

I also was not sure if I fully understood the point of the experiments targeting FtsH1. The authors mention that they performed the first functional study FtsH1, but I think this is an overstatement and this should be reframed as a preliminary functional study. The authors do not mention that the protein was already localized to the apicoplast in 2009 by Karnataki et al. (PMID: 17822404). Then, they characterize the depletion of the protein, which leads to apicoplast loss. Given what we knew of the protein from Plasmodium and Toxoplasma, the results from the authors are not surprising at all. Yet, the results concerning FtsH1 are displayed as a main figure, but the authors never build upon those results again. To me, this portion of the manuscript seems disconnected from the main story (Which can be divided into the strategy to bypass the four output-generating pathways, the generation of the apicoplast-less parasites via actinonin treatment and their subsequent preliminary characterization), so I would recommend to move them to the supplementary material. To me, a functional study will go beyond what the authors did to fully understand the role of this metalloprotease, and will probably be a standalone paper.

We appreciate the reviewer's insightful comments regarding our investigation of FtsH1. We sincerely apologize for the oversight in not citing the 2009 study by Karnataki *et al.* detailing the localization of TgFtsH1¹, which has now been included in our manuscript (see **line 601**).

We acknowledge that our initial characterization of FtsH1 was preliminary and distracting from the focus of this study. To gain deeper insight into the function of FtsH1 beyond the quantitative proteomic analyses following its downregulation, we conducted two additional experiments as part of this revision:

1. A **pull-down experiment** with Ty-tagged FtsH1 to identify potential interactors and substrates.
2. **Total quantitative proteomics** of parasites after 6 hours of actinonin treatment to explore the functional consequences of FtsH1 disruption prior to organelle loss.

Unfortunately, we were unable to identify specific substrates or consequences associated with FtsH1 downregulation (for further details concerning the above experiments, please refer to our response to **Reviewer 2**).

Consequently, and considering this reviewer's feedback that the FtsH1 data appeared disconnected from the main story, we have decided to remove a substantial portion of FtsH1 data. However, some relevant phenotypic analyses are included in **Fig. 8**, providing a contrast with other essential apicoplast proteins. Importantly, this work still represents the first genetic targeting of FtsH1.

While we have removed some data on FtsH1, we have considerably strengthened our investigation of apicoplast-less *T. gondii* parasites. This includes a comprehensive analysis of the apicoplast bypass for identifying apicoplast-targeting drugs, an in-depth examination of the metabolic capacity of apicoplast-less parasites, and novel transcriptomic analyses. The fully revised **Figs. 4-8**, now include numerous new data to support these findings.

Here is a list of other elements that, in my opinion, need to be addressed to improve to quality of the paper.

- Figure 1 is extremely dense and complex, it will be better only highlight the proteins studied by the authors and just mention each of the pathways. We do not need the names and abbreviation of all the enzymes and this stage.

We thank the reviewer for pointing this out. **Fig. 1** has been simplified, only the names of the enzymes studied here are shown now, as suggested.

- Line 290: "its effectiveness (of the MVA cassette) remained unclear". What do the authors mean? Can they clarify this point? In the referenced paper (PMID: 36449552), it is shown that adding MEV to the media rescued the deficiency of the triosephosphate isomerase 2 mutants generated (Figure 6 of the aforementioned paper). That in my mind seems that its effectiveness was demonstrated?

The study by Niu *et al.*, 2022² demonstrated a partial rescue of parasites lacking the apicoplast-localized triosephosphate isomerase 2 (TPI2) through expression of the MVA cassette. However, it is important to note that TPI2 is only indirectly linked to the MEP/DOXP pathway. TPI2 catalyzes the conversion of DHAP to GA3P, which is involved in the synthesis of IPP but also needed to generate reducing power. Notably, GA3P can be imported directly, bypassing the need for TPI2 for the MEP/DOXP pathway altogether. Therefore, the interpretation of the rescue data in their study is less straightforward compared to a direct rescue of an enzyme within the MEP/DOXP pathway, such as IspH, studied here.

Moreover, the rescue of parasites deficient in TPI2 in the cited study is not complete, with plaques developing 3-4 times smaller than under control conditions. In contrast, our findings indicate a complete (100%) rescue of the MEP/DOXP pathway enzyme IspH, upon

expression of the MVA cassette and supplementation of MVL (see **Fig. 3g, h**). Considering this, we argue that our data provides much clearer insights concerning the ability to rescue the MEP/DOXP pathway and the efficiency of this bypass. We have made this point clearer in the revised manuscript (see **lines 699-703**).

- Line 411-412: parasites treated with actinonin formed plaques close to normal size when cultured in ARM. This is an overstatement, the plaques are less than half as big. They are plaques there, but not at all close to the size of the parental strain. Please reword accordingly.

We agree that the original statement overstated the size of the plaques. We have revised this section to clearly acknowledge the reduced size of the plaques following actinonin treatment, which is now accurately reported as 45% of the control size (see **lines 285-288**).

In addition, in our response to **Reviewer 2**, we explicitly recognize the partial rescue of the fatty acid (FA) and heme synthesis pathways, along with the resulting reduction in plaques following total apicoplast loss (see **lines 302-305**).

Beyond this acknowledgment in the results section, we provide a detailed discussion regarding the incompleteness of the rescue and its potential causes. We also speculate on whether further improvements to the bypass conditions (changing the concentration of the supplemented metabolites) could enhance the rescue effect (see **lines 734-746**).

Furthermore, we also discuss in detail the limitations of this bypass as a tool given the reduced fitness (see **lines 734-746**). Specifically, we highlight that this bypass is suitable for numerous approaches, like the ones used in this manuscript (genetics, metabolomics, proteomics transcriptomics, drug assays etc.) but might be less suitable for approaches that rely on equal growth rates (e.g., CRSIPR/Cas9 fitness screens).

- Why did the authors choose to provide complete ARM with FA and LPA even though it seemed that one could be sufficient? They do not provide any reasoning behind that decision (line 421).

When withdrawing either fatty acids (FAs) or lysophosphatidic acid (LPA) from the apicoplast rescue medium (ARM), we did observe a subtle reduction in plaque sizes. These differences were modest compared to the absence and drastic reduction in plaque size when withdrawing mevalonolactone (MVL) or 5-aminolevulinic acid (5ALA), respectively, but significant (see **Supplementary Fig. S4d, e**). Thus, to ensure the best possible growth conditions over the long term, we opted to provide both metabolites. The subtle defect when

withdrawing LPA or FAs was not clearly stated before. This was now added and the reasoning for providing both metabolites in ARM has been provided. This is now stated clearly in the manuscript – see **lines 294-298**.

- Regarding the generation of the apicoplast-less parasites, is this phenotype reversible? It is not clear in the paper if the authors need to keep the parasites in the presence of actinonin and ARM to maintain the phenotype or if the actinonin treatment was only performed for 20 days (line 423).

The apicoplast-less phenotype is not reversible as prolonged disruption of the organelle's functions leads to the loss of its genome (see **Fig. 4l**). Consequently, once lost, the organelle cannot be restored. This is evident from the data showing that parasites devoid of the organelle are unable to survive in regular medium *in vitro* (see **Fig. 4e-h**) and are unable to infect mice (see **Fig. 4n**). We have stated this more explicitly in the revised text – see **lines 337-340**.

Since apicoplast loss is permanent, once established, we removed actinonin from the culture medium of the isolated apicoplast-less clones. Importantly, in the revision we demonstrate that apicoplast loss can occur spontaneously when culturing parasites in apicoplast rescue medium (ARM) without any drug treatment (41% apicoplast loss after 8 passages in ARM). This is shown in the revised **Fig. 4m**.

Minor comments:

- In all the figures with a Western Blot, there is no indication of how many times the WB were repeated in either the legend or in the M&M section.

All western blot experiments, as well as other experiments presented in the paper were independently repeated a minimum of three times (biological replicates) to confirm the reproducibility of the results. This was previously stated in the statistical analyses section of the manuscript but has now also been added to each figure legend.

- In all the figures with an intracellular growth assay, the colors should be the same to avoid any confusion. For instance, in figure 2m choose the same color scheme (blue or orange) instead of blue and orange.

We appreciate the reviewer's perspective on this point; however, we believe this ultimately comes down to a matter of preference. Consistent with several previous studies³⁻⁵, we have chosen to use distinct colors for the control and various mutant strains across all plots,

including intracellular growth assays. While this choice may make differences in cell numbers slightly less obvious, we argue that it facilitates a quicker understanding of the strains and conditions being compared. Overall, this color scheme contributes to a more harmonious design, enhancing the clarity of the manuscript.

- Why do we see several bands in figure S5c? The expression of the MVA cassette did not show that pattern when expressed in the iKD IspH-MVA-HA strain (figure 3h) and the authors do not offer any explanation for this.

We agree with the reviewer that there is a difference between the western blots of these two strains expressing the MVA cassette, which was not acknowledged in the initial submission. Importantly, the MVA polyprotein was consistently (across all 3 biological replicates) detected as a single band in the iKD IspH strain expressing the strain (now **Fig. 3f**), while multiple bands were detected in RH-MVA-HA parasites (now **Supplementary Fig. 4c**). The lower bands below the main band at 113 kDa (predicted molecular weight of the full-size protein) likely correspond to degradation products. We are unsure why this degradation occurs in one strain but not the other. Importantly, however, most of the protein is detected at the expected full size in RH-MVA-HA (**Supplementary Fig. 4c**), localizes to the cytosol by IFA (**Supplementary Fig. 4b**) and the rescue upon MVL supplementation is efficient (**Fig. 4e-h**), indicating that the polyprotein is active. We have now included a comment regarding the differences between the western blots in the two strains and acknowledging the lower bands in the RH protein extract– see **lines 274-278**.

In the same vein, why don't we see both premature and mature forms of Cpn60 in figure 5g like we see in figure 4i for RH parasites?

(We assume that the reviewer refers to the previous **Fig. 4l**, not **4i** – **Fig. 4l** showed a western blot of Cpn60, while **Fig. 4i** showed a plaque assay).

In fact, in the former **Fig. 5g** (now **Fig. 4k**) both, the premature and mature protein can be seen upon closer inspection. In the previous **Fig. 4l**, the accumulation of the premature form became more pronounced only upon downregulation of FtsH1, but in the untreated samples, the premature form of Cpn60 is also a very minor band. Arguably, the premature form is slightly more pronounced in the previous **Fig. 4l**, which can be attributed to modest differences in overall signal intensity and biological variation. Please note that the previous **Fig. 4l** has been removed as part of this revision, aligning with the request to reduce data on FtsH1 and focus on the metabolic bypass. As stated above, **Fig. 5g** has been moved to **Fig. 4k**, as part of the revision of this manuscript.

- Figure 5k: why are the plaque assays in another color? They look very different from the plaque assays from the rest of the figure.

We acknowledge that the plaque assays previously displayed were of suboptimal quality and showed differences in color, due to the varying efficiency of staining. We have replaced these plaques assays with more representative images of higher quality.

Of note, we have opted to remove the data on actinonin and doxycycline. The former was used in this study to trigger apicoplast loss. Hence, differences in its sensitivity were expected. The latter, as pointed out by **Reviewer 2**, also has a substantial impact on translation in the mitochondrion. We chose instead to investigate potential differences between parasites dependent on the apicoplast or not in response to clindamycin (a more specific inhibitor of apicoplast-localized translation) ⁶, atovaquone (control, targeting the mitochondrion) ⁷ and triclosan (an inhibitor of FAS component FabI, with considerable side effects) ⁸. We are convinced that these drugs highlight the differences in sensitivity between apicoplast-dependent and apicoplast-bypassing conditions better and underline the utility of this bypass as a tool to identify apicoplast-targeting drugs. These new plaque assays of higher quality can be found in the completely revised **Fig. 5**.

- Figure S5i: several panels seems to be at a very low res.

As outlined in the previous response, we have replaced the panels of low resolution with images of higher quality - see completely revised **Fig. 5**.

Reviewer #2

In the manuscript titled “Dissecting apicoplast functions through continuous cultivation of *Toxoplasma gondii* devoid of the organelle” by Chen et al. have attempted to expand the knowledge of apicoplast functions beyond the already extensive literature by knocking out the apicoplast completely in the model apicomplexan *Toxoplasma gondii*. The authors achieved apicoplast removal by first sequentially disrupting the four essential metabolic pathways housed by the organelle and then attempting to bypass each pathways either by directly supplying the end products of the pathways in the culture medium or by expressing enzymes that can convert supplemented substrates into the products/outputs of the essential pathways. In each case this work relies on robust knowledge that was already available in the field.

After these validations, the authors chooses to knock out the apicoplast by actinonin treatment, which inhibits the FtsH1 metalloprotease required for apicoplast biogenesis. This apicoplast-knocked-out cell line is maintained in continuous culture by supplementing the medium through the previously known rescue or bypass methods validated at the beginning of the work.

The deletion of the apicoplast has previously been accomplished in the intraerythrocytic stage of the closely related *Plasmodium* species and thus the conceptual progress minor to the field. However, this study presents the first instance of such an undertaking in *Toxoplasma*, which has high potential to be valuable for *Toxoplasma* researchers and to aid in a detailed understanding of the biogenesis and function of this important organelle. Yet, the current work does not present novel findings. The comprehensive analysis of all the cell lines, as well as the quality of data generated by the authors, is commendable. Unfortunately, the findings do not yet add much upon existing knowledge to further elucidate the biology of the apicoplast in *Toxoplasma*.

We appreciate the reviewer’s critical perspective on the novelty of our findings. In response, we would like to highlight the novel contributions of this study:

- **First Report of Apicoplast-Bypass in *T. gondii*:** This is the first demonstration that the apicoplast can be bypassed in *T. gondii* (**Fig. 4**). Indeed, a similar bypass has been described for blood-stage malaria parasites ⁹, where the organelle has a reduced function and only requires a single metabolite (IPP) to survive loss of the organelle. Instead, achieving this in *T. gondii* is more complex due to the organelle’s extensive functions and the challenges in providing metabolites that bypass the heme and

isoprenoid synthesis in this parasite, compounds that are not readily salvaged by these parasites.

- **Utility of Apicoplast-Less *T. gondii*:** As detailed in our manuscript, this bypass has broader implications in *T. gondii* compared to *P. falciparum*, given the organelle's diverse critical roles and the genetic tractability of *T. gondii*. Indeed, we demonstrate that genetic mutants can be readily generated in parasites lacking the organelle (see **Supplementary Fig. 6**) and that apicoplast-less *T. gondii* can be used to identify inhibitors of apicoplast-resident pathways which are dispensable in blood stage *P. falciparum* (see **Fig. 5**). This relatively complex bypass described here can likely be employed to investigate the apicoplast's functions in other stages and Apicomplexa (e.g., *T. gondii* bradyzoites and liver stage *P. falciparum*).
- **Methodical Characterization of Apicoplast Mutants:** We have successfully achieved bypass of the apicoplast through a comprehensive characterization of mutants across the four output-generating pathways and a supporting pathway (**Figs. 2, 3**). Many of the genes examined in this context (e.g., FabG, LipA and PBGD) have, to the best of our knowledge, not been targeted genetically in *T. gondii* before, and a metabolic bypass has not been described for the above genes or for IspH. Additionally, in the context of this study, we describe and phenotype several other mutants of apicoplast-resident proteins, which have not been described before.
- **Expanded Metabolomic Analysis:** We have considerably bolstered the metabolomic analysis of apicoplast-deficient parasites, compared to what was presented in the initial submission. A previous study had investigated the metabolic consequences upon loss of the organelle in *P. falciparum*¹⁰, and identified surprisingly few differences. Here, we decided to specifically investigate the metabolic activity in the remnant vesicles following apicoplast loss (**Fig. 6b-d**). Our findings indicate for the first time that fatty acid synthesis and heme synthesis do not occur or do not contribute to parasite fitness after the loss of the organelle, respectively.
- **First Detailed Proteomic Analysis:** We conducted a thorough proteomic analysis of apicoplast-deficient parasites, finding a specific and significant downregulation of most apicoplast proteins (**Fig. 7a-c**). To our knowledge, this is the first report demonstrating that apicoplast proteins are significantly reduced in parasites lacking the organelle, enabling their identification. This provides a rich resource to identify novel apicoplast proteins.

- **Insights into Apicoplast Loss throughout Evolution:** We report here considerable changes in the proteome and metabolome when transferring parasites from regular medium to medium that bypasses the apicoplast's functions. Additionally, as part of this revision, we report for the first time that the apicoplast is lost frequently and spontaneously, which is only tractable when culturing parasites in apicoplast rescue medium, without the need for a trigger (e.g., a drug disrupting the apicoplast) (**Fig. 4m**). We argue that these observations offer some insights into how the organelle was lost throughout evolution in parasites that established a lifestyle in nutrient-rich conditions.
- **Transcriptomic Analysis:** Our novel transcriptomic analysis (**Fig. 7d, e**) reveals that the reduction in apicoplast proteins is not due to decreased transcription levels of the corresponding genes. A transcriptomic analysis had previously been performed using *P. falciparum* parasites lacking the apicoplast¹⁰. However, that study did not investigate the abundance of proteins upon loss of the organelle, which is a key piece of data in our study. Having both datasets, we can conclude that the decreased abundance of apicoplast proteins following loss of the organelle is not due to their reduced transcription, but rather a consequence of the failed trafficking to the correct destination, presumably destabilizing the mistargeted proteins.
- **Identification of Apicoplast Proteins:** Using our proteomic dataset, we identify putative apicoplast proteins based on their reduced abundance in parasites lacking the organelle. We validate the localization of one putative apicoplast proteins (TGGT1_201270) and identify a novel apicoplast protein (TGGT1_248770) (**Fig. 8**). Furthermore, we provide a basic phenotypic characterization of this newly identified apicoplast protein, demonstrating that it is a membrane protein essential for apicoplast maintenance and parasite survival, and is conserved only within Coccidia.

Overall, we believe these novel findings significantly advance our understanding of the apicoplast organelle and its potential as a drug target, and we hope the reviewer recognizes this contribution.

Major points:

1. While the loss of the apicoplast in the clonally selected *T. gondii*-Apico cell line has been thoroughly validated using immunofluorescence assays (IFA) to stain various apicoplast markers, PCR for genomic fragments, and electron microscopy (EM), the same cannot be asserted for other cell lines used as validation of previous work, where partial or extreme apicoplast loss is claimed. The apicoplast, with its multiple membrane-bound

compartments and resident genome, may not be entirely absent if only one protein (e.g., ATrx1) is missing from a specific compartment. As long as the organelle's genome is present, the organelle itself cannot be considered lost. Therefore, it is recommended that the authors validate apicoplast loss in the various knockdown cell lines using PCR for apicoplast genomic regions and, if possible, with IFA for an apicoplast marker in a different compartment of the organelle.

We appreciate the reviewer's critical feedback. We were initially skeptical about the reliability of quantitative PCR results, given that apicoplast loss is partial and the genome typically exists in 15-25 copies per apicoplast^{11,12}. Therefore, we instead opted to use an additional marker, Cpn60 (an apicoplast lumen marker), alongside the membrane marker Atrx1 and DAPI to stain the apicoplast genome.

For two mutants, FabG and the newly identified apicoplast protein TGGT1_248770, our results show that quantifying partial apicoplast loss yields highly consistent results, whether based on Cpn60, Atrx1, or DAPI staining (see **Supplementary Fig. 2g, h** and **Fig. 8h**). Notably, DAPI staining was consistently absent in parasites that were negative for Cpn60 (see **Supplementary Fig. 2g** and **Fig. 8h**), strongly indicating that the genome is indeed lost in the parasites classified as apicoplast-less based on the protein markers.

As the reviewer highlights, we have made significant efforts to validate the complete apicoplast loss in the *T. gondii*^{-Apico} line. We went to these lengths since this marks the first report of viable *T. gondii*^{-Apico}. In contrast, partial apicoplast loss, such as that observed in FabG, has been well-documented in mutants of the FASII pathway, and was often quantified using a single apicoplast marker^{4,13}.

2. For iKD FabG, ATS1, and LipA, the authors claim that they are bypassing the essential role of these enzymes by supplementing the medium with the respective end products. However, according to plaque assay results, growth defects are only partially rescued; it is also unclear whether these cell lines with supplemented medium can be grown indefinitely in culture. With these points in mind, it would be wrong to claim that these pathways have been bypassed—rather, in this case, the defect has simply been partially rescued. It is notable that this rescue effect for both FA and lipid precursor synthesis and accessory pathways is not a new result and has already been shown in several publications before, as mentioned in the manuscript.

We recognize that it was not clearly acknowledged in the initial submission that our metabolite supplementation of exogenous fatty acids and lysophosphatidic acid presents a

partial rescue rather than a complete bypass. This is now clearly stated/ corrected in the text – see for instance **lines 86-87, 96, 147, 174, 176, 217, 237, 287 and 735**. We also discuss that this partial rescue observed for some pathways likely contributes to the impaired fitness of parasite lacking the organelle compared to their controls see **lines 285-288**.

Concerning the question whether these can be sustained indefinitely in culture: we did not assess this here specifically for these inducible strains (iKD FabG, LipA, ATS1). However, there are two lines of evidence which very strongly suggest that this is the case:

- 1) Our study details the continuous culture (>60 passage) of apicoplast-less *T. gondii*. These parasites lack the entire organelle and likely the activity of all its associated pathways (see new **Fig. 6b-d**). Since parasites that lack the entire organelle and its resident pathways can be sustained indefinitely in apicoplast rescue medium, it appears sensible to assume that the mutants mentioned by the reviewer (iKD FabG, LipA, ATS1) can be equally sustained.
- 2) For several components of the FASII and related lipid synthesis pathway, stable knock out (KO) mutants have been generated and characterized (Δ FabZ, Δ FabD, Δ PDH)^{4,14}, which can be maintained in culture indefinitely, and present improved growth and fitness in presence of exogenous fatty acids.

3. In the third results section, the authors aim to further characterize the function of FtsH1 in *T. gondii*. FtsH1 has been previously shown to localize in the apicoplast, is important for apicoplast biogenesis in Plasmodium and Toxoplasma, and is predicted to be involved in membrane protein processing in the apicoplast. Although the authors performed mass spectrometry analysis on the FtsH1-depleted cell line, they did not elucidate the protein's function beyond what was already known. Furthermore, there is insufficient explanation or analysis regarding the upregulation or downregulation of certain proteins, particularly why most of these are ER-resident proteins.

→ Please see response to Reviewer 2, Major Point 5 (see page 15)

4. Can the authors confirm that some of these upregulated proteins upon FtsH1-depletion are substrates of FtsH1 through biochemical analyses? Can it be verified that the proteins accumulating in the iKD-FtsH1-Ty cell lines have premature peptides or have processing defects?

→ Please see response to Reviewer 2, Major Point 5 (see page 15)

5. Additionally, the statement about the interaction between Cpn60 and FtsH1 is quite handwavy without further experimental evidence. While the authors compare this to plants, it should be noted that, unlike *Toxoplasma*, which has only one FtsH protein, plants can

contain several FtsH proteins—12 in case of Arabidopsis. Only one of these, FtsH11, localizes in the inner envelope membrane, has been shown to interact with Cpn60 in its proteolytically inactive mutant form, questioning the biological relevance of this "artefactual" interaction. Comparing this to justify the downregulation of Cpn60 in Toxoplasma seems tenuous without further exploration and evidence. The authors could perform immunoprecipitation of the FtsH1-Ty tagged cell line to identify the interaction partners of FtsH1. Likewise, are the five downregulated proteins all interacting partners of FtsH1?

We acknowledge that our analysis of FtsH1 was preliminary and did not provide clear novel insights concerning the function and substrates of FtsH1. We also recognize that the results from our proteomic analysis, following the downregulation of FtsH1, were not sufficiently discussed and that a potential interaction between FtsH1 and Cpn60 was speculative.

To obtain novel insights into the function of FtsH1, beyond the quantitative proteomic analyses following downregulation of FtsH1 (see original submission), we performed two additional experiments as part of this revision:

- 1) a **pull-down experiment** of Ty-tagged-FtsH1 – to identify potential interactors/substrates.
- 2) **total quantitative proteomics** of parasites after 6 hours of actinonin treatment - to obtain insights into the function of FtsH1 (attempting to observe consequences of FtsH1 disruption, prior to loss of the organelle).

The results from the above analyses are briefly summarized here:

The top hit from a pull-down of FtsH1-Ty analysis was FtsH1 itself. For only two other proteins, significantly more peptides were detected in the pulldown of FtsH1-Ty parasites compared to the control (DiCre). These were, however, not associated with the apicoplast and for each, only few peptides were detected per replicate (Data not shown).

Table 2. Total quantitative proteomics following 6 hours of actinonin treatment (DiCre parasites treated, compared to untreated). Only significantly upregulated proteins are shown (p-value < 0.05, FC > 1.5). Apicoplast proteins are highlighted in green.

Gene ID	Product Description	Fold-change (AVG Log2 Ratio)	Localisation	T. gondii GT1 CRISPR Phenotype
Upregulated Proteins				
TGGT1_246600	ABC1 family protein	5.1937	ER	-0.49
TGGT1_236920	hypothetical protein	1.1864	nucleus - non-chromatin	-1.16
TGGT1_206500	hypothetical protein	1.0510	mitochondrion - soluble	-4.14
TGGT1_313900	non-specific serine/threonine protein kinase	0.8371	NA	0.01
TGGT1_206340	hypothetical protein	0.8318	Golgi	1.95
TGGT1_217550	hypothetical protein	0.8314	cytosol	2.07
TGGT1_226780	zinc finger, C3HC4 type (RING finger) domain-containing protein	0.8088	nucleus - chromatin	-0.15
TGGT1_313540	hypothetical protein	0.7275	ER	-3.47
TGGT1_293260	ATPase/histidine kinase/DNAgyraseB/HSP90 domain-cont. protein	0.7067	NA	0.02
TGGT1_223125	ubiquitin family protein	0.6852	apicoplast	-4.91
TGGT1_223126	hypothetical protein	0.6688	Golgi	-3.88
TGGT1_223127	NLI interacting factor family phosphatase	0.6625	NA	-3.89
TGGT1_223128	ribosomal protein RPS27	0.6409	nucleus - chromatin	-3
TGGT1_223129	tetratricopeptide repeat-containing protein	0.6342	NA	1.37
TGGT1_223130	heat shock factor binding protein 1 protein	0.6199	NA	0.89
TGGT1_223131	hypothetical protein	0.6123	apicoplast	-1.81
TGGT1_223132	Tyrosine kinase-like (TKL) protein	0.6086	nucleus - chromatin	-5.18
TGGT1_223133	BolA family protein	0.6055	cytosol	2.11
TGGT1_223134	DnaJ domain-containing protein	0.5959	NA	-3.81
TGGT1_223135	hypothetical protein	0.5957	NA	1.21
TGGT1_223136	hypothetical protein	0.5937	NA	1.36
TGGT1_223137	hypothetical protein	0.5914	NA	-2.84
TGGT1_223138	putative calmodulin	0.5826	IMC	0.11

Amongst 23 significantly upregulated proteins upon 6 hours actinonin treatment, we found two putative apicoplast proteins. Instead, 109 proteins were significantly downregulated following actinonin treatment (results not shown), but included no apicoplast proteins.

In summary, despite considerable efforts (total quantitative proteomics following FtsH1 downregulation, total quantitative proteomics following short-term actinonin treatment and pull-down of FtsH1), we were unable to obtain clear meaningful insights into the function and substrates of FtsH1 beyond what was previously known.

Due to this, and the remark from **Reviewer 1**, highlighting that this data is disconnected from the rest of the manuscript and distracting from the main findings, we have decided to remove it. Some of the relevant phenotyping data of FtsH1 are now included in the revised **Fig. 8** in comparison with other, newly identified, essential apicoplast proteins. Notably, this study still provides the first study, targeting FtsH1 genetically.

Importantly, while removing this data, we have significantly bolstered the investigation of apicoplast-less *T. gondii* parasites, including a detailed analysis of the utility of the apicoplast bypass to identify apicoplast-targeting drugs in *T. gondii*, an in-depth exploration of the metabolic capacity of apicoplast-less parasites and novel transcriptomic analyses (See fully revised **Figs. 4-8**, containing numerous new pieces of data).

6. A potentially exciting outcome of maintaining an apicoplast-less *T. gondii* cell line would be to test whether inhibitors/compounds have specific targets in the apicoplast, as also pointed out by the authors. In this case, the authors tested doxycycline to verify their claims, as it should target the ribosome in the apicoplast. However, there is literature that suggests doxycycline also targets mitochondrial ribosomes. It would be helpful if the authors chose an inhibitor which is not actinonin that was used to induce apicoplast loss, but for which the target is known to be exclusive to the apicoplast (e.g., clindamycin), meaning the parasites would be completely insensitive to its use in culture.

We agree with the reviewer that the previously tested drugs were suboptimal. Instead of testing the sensitivity to actinonin and doxycycline in parasites dependent on the apicoplast or not, we now show data using clindamycin (a more specific inhibitor of apicoplast-localized translation, following the reviewer's advice)⁶, atovaquone (control, targeting the mitochondria)⁷ and triclosan (an inhibitor of FAS component FabI, with considerable side effects)⁸. This data is presented in a completely revised **Fig. 5**. This new data reveals a dramatic shift in the sensitivity to clindamycin under apicoplast bypass conditions and shows a difference in sensitivity to triclosan. The latter is an inhibitor of FabI of the FASII pathway (with significant off target effects). We used triclosan to highlight that apicoplast-less *T. gondii* can be used to identify drugs targeting the apicoplast that could not be identified using the bypass in blood stage *P. falciparum*, since fatty acid synthesis is dispensable in this stage.

7. As the apicoplast is known to be a metabolic hub for the parasite, it is disappointing to see that the authors haven't provided an in-depth analysis of the major changes in metabolites using the metabolomics data. As presented the data is inconclusive or hard to interpret. This seems like a missed opportunity to delve into insights that the field does not already hold about apoplast function.

We agree with the reviewer that the metabolomic data was not presented prominently and discussed thoroughly in the previous submission. In response, we have revised the presentation of the data and moved it to the main **Fig. 6a**, where they are presented as a heatmap. We now discuss in detail the metabolomic differences between parasites with and without an apicoplast and find that these differences are largely abolished when culturing parasites in apicoplast rescue medium. Notably, a previous study had also investigated the metabolism of *P. falciparum* lacking the organelle and had found relatively few changes¹⁰.

To go beyond this, we have devised and performed analyses that specifically test whether metabolically activity can, to some extent, resume in the remnant vesicles following loss of the organelle. We employed stable isotope labelling of purified parasites and generated a KO

mutant deficient of a heme synthesis enzyme, uroporphyrinogen III synthase (UROS), in a line devoid of the apicoplast. Using these approaches, we clearly reveal for the first time that fatty acid synthesis and heme synthesis do not occur or do not contribute to parasite fitness after the loss of the organelle, respectively (**Fig. 6b-d**). We believe that these changes significantly improve the presentation of the metabolomic data and provide additional insights.

Minor points:

1. Line 897, “plague” should be “plaque”

This has been corrected (now **line 902**).

2. Line 417 and Suppl. Fig 5d – “MVL” instead of “MEV”

This has been corrected (now **Supplementary Fig. 4d**).

Reviewer #3

We thank all reviewers for their critical input, helping to improve the structure and overall impact of this manuscript.

Please note additionally, that **Olivier von Rohr** was added to the list of authors, due to his contributions during this revision.

Furthermore, the raw data corresponding to the newly generated data sets (transcriptomics and ¹³C-labelling analyses of fatty acids) have been deposited and can be accessed through the links provided in the manuscript.

- 1 Karnataki, A., DeRocher, A. E., Feagin, J. E. & Parsons, M. Sequential processing of the *Toxoplasma* apicoplast membrane protein FtsH1 in topologically distinct domains during intracellular trafficking. *Mol Biochem Parasitol* **166**, 126-133 (2009). <https://doi.org/10.1016/j.molbiopara.2009.03.004>
- 2 Niu, Z. *et al.* Two apicoplast dwelling glycolytic enzymes provide key substrates for metabolic pathways in the apicoplast and are critical for *Toxoplasma* growth. *PLoS Pathog* **18**, e1011009 (2022). <https://doi.org/10.1371/journal.ppat.1011009>
- 3 Kloehn, J., Lunghi, M., Varesio, E., Dubois, D. & Soldati-Favre, D. Untargeted Metabolomics Uncovers the Essential Lysine Transporter in *Toxoplasma gondii*. *Metabolites* **11** (2021). <https://doi.org/10.3390/metabo11080476>
- 4 Krishnan, A. *et al.* Functional and Computational Genomics Reveal Unprecedented Flexibility in Stage-Specific *Toxoplasma* Metabolism. *Cell Host Microbe* **27**, 290-306 e211 (2020). <https://doi.org/10.1016/j.chom.2020.01.002>
- 5 Lunghi, M. *et al.* Pantothenate biosynthesis is critical for chronic infection by the neurotropic parasite *Toxoplasma gondii*. *Nat Commun* **13**, 345 (2022). <https://doi.org/10.1038/s41467-022-27996-4>
- 6 Camps, M., Arrizabalaga, G. & Boothroyd, J. An rRNA mutation identifies the apicoplast as the target for clindamycin in *Toxoplasma gondii*. *Mol Microbiol* **43**, 1309-1318 (2002). <https://doi.org/10.1046/j.1365-2958.2002.02825.x>
- 7 Siregar, J. E. *et al.* Direct evidence for the atovaquone action on the *Plasmodium* cytochrome bc1 complex. *Parasitol Int* **64**, 295-300 (2015). <https://doi.org/10.1016/j.parint.2014.09.011>
- 8 Shears, M. J., Botte, C. Y. & McFadden, G. I. Fatty acid metabolism in the *Plasmodium* apicoplast: Drugs, doubts and knockouts. *Mol Biochem Parasitol* **199**, 34-50 (2015). <https://doi.org/10.1016/j.molbiopara.2015.03.004>
- 9 Yeh, E. & DeRisi, J. L. Chemical rescue of malaria parasites lacking an apicoplast defines organelle function in blood-stage *Plasmodium falciparum*. *PLoS Biol* **9**, e1001138 (2011). <https://doi.org/10.1371/journal.pbio.1001138>
- 10 Swift, R. P. *et al.* A mevalonate bypass system facilitates elucidation of plastid biology in malaria parasites. *PLoS Pathog* **16**, e1008316 (2020). <https://doi.org/10.1371/journal.ppat.1008316>
- 11 Matsuzaki, M., Kikuchi, T., Kita, K., Kojima, S. & Kuroiwa, T. Large amounts of apicoplast nucleoid DNA and its segregation in *Toxoplasma gondii*. *Protoplasma* **218**, 180-191 (2001). <https://doi.org/10.1007/BF01306607>
- 12 Reiff, S. B., Vaishnav, S. & Striepen, B. The HU protein is important for apicoplast genome maintenance and inheritance in *Toxoplasma gondii*. *Eukaryot Cell* **11**, 905-915 (2012). <https://doi.org/10.1128/EC.00029-12>
- 13 Li, Z. H., Ramakrishnan, S., Striepen, B. & Moreno, S. N. *Toxoplasma gondii* relies on both host and parasite isoprenoids and can be rendered sensitive to atorvastatin. *PLoS Pathog* **9**, e1003665 (2013). <https://doi.org/10.1371/journal.ppat.1003665>
- 14 Liang, X. *et al.* Acquisition of exogenous fatty acids renders apicoplast-based biosynthesis dispensable in tachyzoites of *Toxoplasma*. *J Biol Chem* **295**, 7743-7752 (2020). <https://doi.org/10.1074/jbc.RA120.013004>

Point-by-Point Response to Reviewers

We thank the reviewers for their valuable feedback and appreciate the opportunity to revise our manuscript accordingly. The reviewers only had minor suggestions/ requests at this stage, which we have addressed as outlined below.

Reviewer #1 (Remarks to the Author)

The authors have greatly improved the quality of the manuscript. They have addressed all my comments, and by restructuring the paper they have enhanced its readability and focus. Congratulations!

→ We thank reviewer 1 for the attention to detail and critical comments, which have helped to improve the manuscript considerably.

I have only a few minor comments that can be solved before publication:

1) Typo in line 347: "in" is repeated two times

→ This has now been corrected – see **line 1755** (note that all line numbers provided here refer to the manuscript with tracked changes).

2) The generation of the RH-MVA-HA–Apico Δ UROS lacks one detail. This gene is essential in normal culture conditions, so where all the experiments to generate the strain performed in ARM media? I am assuming so since the strain is not viable in NM for extended periods of time, but it should be mentioned in the M&M section. Of note, it is interesting to see how an essential gene was now being treated as an indispensable gene (by simply replacing it with an antibiotic resistant cassette) in the RH-MVA-HA–Apico background.

→ We acknowledge that details regarding this strain were missing in the previous manuscript. We have now added a detailed description. In brief: yes, the mutant parasites lacking UROS were drug selected and cloned in ARM. Of note, the apicoplast loss occurred spontaneously during these steps, further consolidating the spontaneous apicoplast loss observed under permissive conditions (culture in ARM). *I.e.*, the transfection to delete UROS was not performed on RH-MVA-HA^{-Apico}, but on RH-MVA-HA parasites, which lost their apicoplast spontaneously, resulting in RH-MVA-HA^{-Apico} Δ UROS parasites. This has now been corrected and is described clearly in lines see **line 359 – 367**.

3) Line 627: TGGT1_248770 has a hydrophobic region near its C terminus. The authors should mention how they found this region. Was it by using data available on ToxoDB? Or via other means?

→ We thank the reviewer for pointing this out. This has now been added – see **lines 483 – 486**. In fact, transmembrane domains were predicted by some prediction tools, but not by others. The solubility assay clearly confirms the presence of one or more transmembrane domains.

4) I believe the solubility assay protocol is missing from the Materials and Method Section. I was able to find a protocol for all the other procedures in the manuscript except for that one.

- We thank the reviewer for pointing this out. The details for this assay have now been added to the description of Western blots in the material and methods section – see **lines 749 – 753**.

Reviewer #2 (Remarks to the Author)

In this revised version of the manuscript, the authors have responded satisfactorily to all our previous queries. Therefore, I recommend this article to be accepted for publication.

- We thank reviewer 2 for the feedback and help to improve the manuscript.

Reviewer #3 (Remarks to the Author)

- We thank reviewer 3 for investing the time to co-review this manuscript.

Reviewer #4 (Remarks to the Author)

I was asked by the editor to focus on the transcriptomic data presented in this manuscript (Specifically, the data presented in Fig. 7d, and supplementary data included in Table 6) and hence my comments are largely focused on this section of the manuscript.

The authors had previously looked at the protein levels in parasites with (RH-MVA-HA) and without the apicoplast (RH-MVA-HA-Apico). The data from these experiments had indicated that nuclear encoded apicoplast protein levels were downregulated in parasites lacking apicoplast. To determine if the downregulation of proteins is due to regulation at the transcriptional level or degradation in the absence of proper trafficking, authors performed RNA seq analysis.

The data presented in Fig.7d is compiled from three experiments is robust and the authors correctly interpret the data. Since most of the proteins that were found to be downregulated at the protein level, did not show significant changes in their RNA transcripts number suggests that the decreased abundance is mainly due to impaired stability. Hence the authors interpretation is indeed spot on.

Additionally, after careful reading of the revised manuscript, I feel that authors have adequately addressed all the reviewers comments.

- We thank reviewer 4 for the specific feedback concerning the transcriptomic data as well

as the overall assessment.

Other changes to the manuscript:

- We have edited **Figure 5**, to include an additional piece of data (intracellular growth assay during drug treatment), which supports the previously presented data (plaque assay during drug treatment). The results section and figure legend have been updated accordingly. This minor change was approved by the Editor in an email exchange on 30.01.2025.
- We have edited a funding source in the acknowledgements, which was previously omitted – see **line 1670**.
- We have added a paragraph in the discussion pointing out that the loss of the apicoplast leads to the downregulation of several proteins of complex III and IV of the mitochondrial respiratory chain. We briefly discuss the possible causes for this – see **lines 609 - 636**.
- We have updated the Methods description for the metabolomic analyses, restructuring it for enhanced clarity and providing some additional information – see **lines 839 - 900**.
- All other changes are based on the requests regarding the formatting (Checklist for authors)